# FusionNet: Fusing via Fully-Aware Attention with Application to Machine Comprehension

**Hsin-Yuan Huang\*[1,2], Chenguang Zhu[1], Yelong Shen[1], Weizhu Chen[1]**
[1]Microsoft Business AI and Research
[2]National Taiwan University
momohuang@gmail.com, {chezhu,yeshen,wzchen}@microsoft.com

## Abstract

This paper introduces a new neural structure called FusionNet, which extends existing attention approaches from three perspectives. First, it puts forward a novel concept of "history of word" to characterize attention information from the lowest word-level embedding up to the highest semantic-level representation. Second, it identifies an attention scoring function that better utilizes the "history of word" concept. Third, it proposes a fully-aware multi-level attention mechanism to capture the complete information in one text (such as a question) and exploit it in its counterpart (such as context or passage) layer by layer. We apply FusionNet to the Stanford Question Answering Dataset (SQuAD) and it achieves the first position for both single and ensemble model on the official SQuAD leaderboard at the time of writing (Oct. 4th, 2017). Meanwhile, we verify the generalization of FusionNet with two adversarial SQuAD datasets and it sets up the new state-of-the-art on both datasets: on AddSent, FusionNet increases the best F1 metric from 46.6% to 51.4%; on AddOneSent, FusionNet boosts the best F1 metric from 56.0% to 60.7%.

## 1 Introduction

Teaching machines to read, process and comprehend text and then answer questions is one of key problems in artificial intelligence. Figure 1 gives an example of the machine reading comprehension task. It feeds a machine with a piece of context and a question and teaches it to find a correct answer to the question. This requires the machine to possess high capabilities in comprehension, inference and reasoning. This is considered a challenging task in artificial intelligence and has already attracted numerous research efforts from the neural network and natural language processing communities. Many neural network models have been proposed for this challenge and they generally frame this problem as a machine reading comprehension (MRC) task (Hochreiter & Schmidhuber, 1997; Wang et al., 2017; Seo et al., 2017; Shen et al., 2017; Xiong et al., 2017; Weissenborn et al., 2017; Chen et al., 2017a).

**Context:** The Alpine Rhine is part of the Rhine, a famous European river. The Alpine Rhine begins in the most western part of the Swiss canton of Graubünden, and later forms the border between Switzerland to the West and **Liechtenstein** and later Austria to the East. On the other hand, the Danube separates Romania and Bulgaria.

**Question:** What is the other country the Rhine separates Switzerland to?

**Answer:** **Liechtenstein**

Figure 1: Question-answer pair for a passage discussing Alpine Rhine.

The key innovation in recent models lies in how to ingest information in the question and characterize it in the context, in order to provide an accurate answer to the question. This is often modeled as attention in the neural network community, which is a mechanism to attend the question into the context so as to find the answer related to the question. Some (Chen et al., 2017a; Weissenborn et al., 2017) attend the word-level embedding from the question to context, while some (Wang et al., 2017) attend the high-level representation in the question to augment the context. However we observed

---

\*Most of the work was done during internship at Microsoft, Redmond.

that none of the existing approaches has captured the full information in the context or the question, which could be vital for complete information comprehension. Taking image recognition as an example, information in various levels of representations can capture different aspects of details in an image: pixel, stroke and shape. We argue that this hypothesis also holds in language understanding and MRC. In other words, an approach that utilizes all the information from the word embedding level up to the highest level representation would be substantially beneficial for understanding both the question and the context, hence yielding more accurate answers.

However, the ability to consider all layers of representation is often limited by the difficulty to make the neural model learn well, as model complexity will surge beyond capacity. We conjectured this is why previous literature tailored their models to only consider partial information. To alleviate this challenge, we identify an attention scoring function utilizing all layers of representation with less training burden. This leads to an attention that thoroughly captures the complete information between the question and the context. With this fully-aware attention, we put forward a multi-level attention mechanism to understand the information in the question, and exploit it *layer by layer* on the context side. All of these innovations are integrated into a new end-to-end structure called FusionNet in Figure 4, with details described in Section 3.

We submitted FusionNet to SQuAD (Rajpurkar et al., 2016), a machine reading comprehension dataset. At the time of writing (Oct. 4th, 2017), our model ranked in the first place in both single model and ensemble model categories. The ensemble model achieves an exact match (EM) score of 78.8% and F1 score of 85.9%. Furthermore, we have tested FusionNet against adversarial SQuAD datasets (Jia & Liang, 2017). Results show that FusionNet outperforms existing state-of-the-art architectures in both datasets: on AddSent, FusionNet increases the best F1 metric from 46.6% to 51.4%; on AddOneSent, FusionNet boosts the best F1 metric from 56.0% to 60.7%. In Appendix D, we also applied to natural language inference task and shown decent improvement. This demonstrated the exceptional performance of FusionNet. An open-source implementation of FusionNet can be found at `https://github.com/momohuang/FusionNet-NLI`.

## 2 MACHINE COMPREHENSION & FULLY-AWARE ATTENTION

In this section, we briefly introduce the task of machine comprehension as well as a conceptual architecture that summarizes recent advances in machine reading comprehension. Then, we introduce a novel concept called history-of-word. History-of-word can capture different levels of contextual information to fully understand the text. Finally, a light-weight implementation for history-of-word, Fully-Aware Attention, is proposed.

### 2.1 TASK DESCRIPTION

In machine comprehension, given a context and a question, the machine needs to read and understand the context, and then find the answer to the question. The context is described as a sequence of word tokens: $C = \{w_1^C, \ldots, w_m^C\}$, and the question as: $Q = \{w_1^Q, \ldots, w_n^Q\}$, where $m$ is the number of words in the context, and $n$ is the number of words in the question. In general, $m \gg n$. The answer **Ans** can have different forms depending on the task. In the SQuAD dataset (Rajpurkar et al., 2016), the answer **Ans** is guaranteed to be a contiguous span in the context $C$, e.g., $\mathbf{Ans} = \{w_i^C, \ldots, w_{i+k}^C\}$, where $k$ is the number of words in the answer and $k \leq m$.

### 2.2 CONCEPTUAL ARCHITECTURE FOR MACHINE READING COMPREHENSION

In all state-of-the-art architectures for machine reading comprehension, a recurring pattern is the following process. Given two sets of vectors, A and B, we enhance or modify *every single vector* in set A with the information from set B. We call this a fusion process, where set B is fused into set A. Fusion processes are commonly based on attention (Bahdanau et al., 2015), but some are not. Major improvements in recent MRC work lie in how the fusion process is designed.

A conceptual architecture illustrating state-of-the-art architectures is shown in Figure 2, which consists of three components.

- Input vectors: Embedding vectors for each word in the context and the question.

| Architectures | (1) | (2) | (2') | (3) | (3') |
|---|---|---|---|---|---|
| Match-LSTM (Wang & Jiang, 2016) | | ✓ | | | |
| DCN (Xiong et al., 2017) | | ✓ | | | ✓ |
| FastQA (Weissenborn et al., 2017) | ✓ | | | | |
| FastQAExt (Weissenborn et al., 2017) | ✓ | ✓ | | | ✓ |
| BiDAF (Seo et al., 2017) | | ✓ | | | ✓ |
| RaSoR (Lee et al., 2016) | | ✓ | ✓ | | |
| DrQA (Chen et al., 2017a) | ✓ | | | | |
| MPCM (Wang et al., 2016) | ✓ | ✓ | | | |
| Mnemonic Reader (Hu et al., 2017) | ✓ | ✓ | | ✓ | |
| R-net (Wang et al., 2017) | | ✓ | | ✓ | |

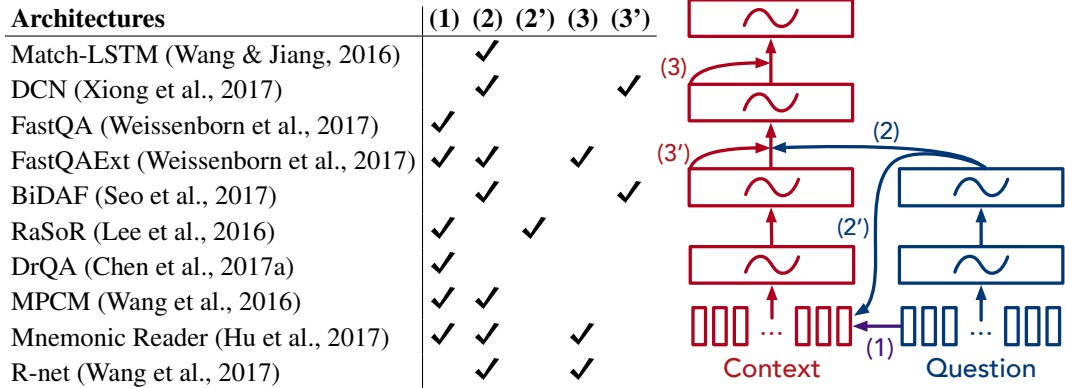

Table 1: A summarized view on the fusion processes used in several state-of-the-art architectures.

Figure 2: A conceptual architecture illustrating recent advances in MRC.

- Integration components: The rectangular box. It is usually implemented using an RNN such as an LSTM (Hochreiter & Schmidhuber, 1997) or a GRU (Cho et al., 2014).
- Fusion processes: The numbered arrows (1), (2), (2'), (3), (3'). The set pointing outward is fused into the set being pointed to.

There are three main types of fusion processes in recent advanced architectures. Table 1 shows what fusion processes are used in different state-of-the-art architectures. We now discuss them in detail.

**(1) Word-level fusion.** By providing the direct word information in question to the context, we can quickly zoom in to more related regions in the context. However, it may not be helpful if a word has different semantic meaning based on the context. Many word-level fusions are not based on attention, e.g., (Hu et al., 2017; Chen et al., 2017a) appends binary features to context words, indicating whether each context word appears in the question.

**(2) High-level fusion.** Informing the context about the semantic information in the question could help us find the correct answer. But high-level information is more imprecise than word information, which may cause models to be less aware of details.

**(2') High-level fusion (Alternative).** Similarly, we could also fuse high-level concept of $Q$ into the word-level of $C$.

**(3) Self-boosted fusion.** Since the context can be long and distant parts of text may rely on each other to fully understand the content, recent advances have proposed to fuse the context into itself. As the context contains excessive information, one common choice is to perform self-boosted fusion after fusing the question $Q$. This allows us to be more aware of the regions related to the question.

**(3') Self-boosted fusion (Alternative).** Another choice is to directly condition the self-boosted fusion process on the question $Q$, such as the coattention mechanism proposed in (Xiong et al., 2017). Then we can perform self-boosted fusion before fusing question information.

A common trait of existing fusion mechanisms is that none of them employs all levels of representation jointly. In the following, we claim that employing all levels of representation is crucial to achieving better text understanding.

## 2.3   FULLY-AWARE ATTENTION ON HISTORY OF WORD

Consider the illustration shown in Figure 3. As we read through the context, each input word will gradually transform into a more abstract representation, e.g., from low-level to high-level concepts. Altogether, they form the history of each word in our mental flow. For a human, we utilize the history-of-word so frequently but we often neglect its importance. For example, to answer the question in Figure 3 correctly, we need to focus on both the high-level concept of *forms the border* and the word-level information of *Alpine Rhine*. If we focus only on the high-level concepts, we will

**Context:** The Alpine Rhine is part of the Rhine, a famous European river. The **Alpine Rhine** begins in the most western part of the Swiss canton of Graubünden, and later **forms the border** between Switzerland to the West and **Liechtenstein** and later Austria to the East. On the other hand, the **Danube** separates Romania and Bulgaria.

**Question:** What is the other country the Rhine separates Switzerland to?

**Answer: Liechtenstein**

``History of Word" Concept

| Input Word | Low level | High level |
|---|---|---|
| Alpine Rhine ⟶ | European river ⟶ | Separating River |
| Forms the border⟶ | Border countries ⟶ | Separates |
| Liechtenstein ⟶ | Country ⟶ | Country, separate |
| Danube ⟶ | European river ⟶ | Separating River |

Figure 3: Illustrations of the history-of-word for the example shown in Figure 1. Utilizing the entire history-of-word is crucial for the full understanding of the context.

confuse *Alpine Rhine* with *Danube* since both are European rivers that separate countries. Therefore we hypothesize that the entire history-of-word is important to fully understand the text.

In neural architectures, we define the history of the $i$-th word, $\mathrm{HoW}_i$, to be the concatenation of all the representations generated for this word. This may include word embedding, multiple intermediate and output hidden vectors in RNN, and corresponding representation vectors in any further layers. To incorporate history-of-word into a wide range of neural models, we present a lightweight implementation we call *Fully-Aware Attention*.

Attention can be applied to different scenarios. To be more conclusive, we focus on attention applied to fusing information from one body to another. Consider two sets of hidden vectors for words in text bodies A and B: $\{\boldsymbol{h}_1^A, \ldots, \boldsymbol{h}_m^A\}$, $\{\boldsymbol{h}_1^B, \ldots, \boldsymbol{h}_n^B\} \subset \mathbb{R}^d$. Their associated history-of-word are,
$$\{\mathrm{HoW}_1^A, \ldots, \mathrm{HoW}_m^A\}, \ \{\mathrm{HoW}_1^B, \ldots, \mathrm{HoW}_n^B\} \subset \mathbb{R}^{d_h},$$
where $d_h \gg d$. Fusing body B to body A via standard attention means for every $\boldsymbol{h}_i^A$ in body A,

1. Compute an attention score $S_{ij} = S(\boldsymbol{h}_i^A, \boldsymbol{h}_j^B) \in \mathbb{R}$ for each $\boldsymbol{h}_j^B$ in body B.
2. Form the attention weight $\alpha_{ij}$ through softmax: $\alpha_{ij} = \exp(S_{ij})/\sum_k \exp(S_{ik})$.
3. Concatenate $\boldsymbol{h}_i^A$ with the summarized information, $\hat{\boldsymbol{h}}_i^A = \sum_j \alpha_{ij} \boldsymbol{h}_j^B$.

In fully-aware attention, we replace attention score computation with the history-of-word.
$$S(\boldsymbol{h}_i^A, \boldsymbol{h}_j^B) \implies S(\mathrm{HoW}_i^A, \mathrm{HoW}_j^B).$$
This allows us to be fully aware of the complete understanding of each word. The ablation study in Section 4.4 demonstrates that this lightweight enhancement offers a decent improvement in performance.

To fully utilize history-of-word in attention, we need a suitable attention scoring function $S(\boldsymbol{x}, \boldsymbol{y})$. A commonly used function is multiplicative attention (Britz et al., 2017): $\boldsymbol{x}^T U^T V \boldsymbol{y}$, leading to
$$S_{ij} = (\mathrm{HoW}_i^A)^T U^T V (\mathrm{HoW}_j^B),$$
where $U, V \in \mathbb{R}^{k \times d_h}$, and $k$ is the attention hidden size. However, we suspect that two large matrices interacting directly will make the neural model harder to train. Therefore we propose to constrain the matrix $U^T V$ to be symmetric. A symmetric matrix can always be decomposed into $U^T D U$, thus
$$S_{ij} = (\mathrm{HoW}_i^A)^T U^T D U (\mathrm{HoW}_j^B),$$
where $U \in \mathbb{R}^{k \times d_h}$, $D \in \mathbb{R}^{k \times k}$ and $D$ is a diagonal matrix. The symmetric form retains the ability to give high attention score between dissimilar $\mathrm{HoW}_i^A, \mathrm{HoW}_j^B$. Additionally, we marry nonlinearity with the symmetric form to provide richer interaction among different parts of the history-of-word. The final formulation for attention score is
$$S_{ij} = f(U(\mathrm{HoW}_i^A))^T D f(U(\mathrm{HoW}_j^B)),$$
where $f(x)$ is an activation function applied element-wise. In the following context, we employ $f(x) = \max(0, x)$. A detailed ablation study in Section 4 demonstrates its advantage over many alternatives.

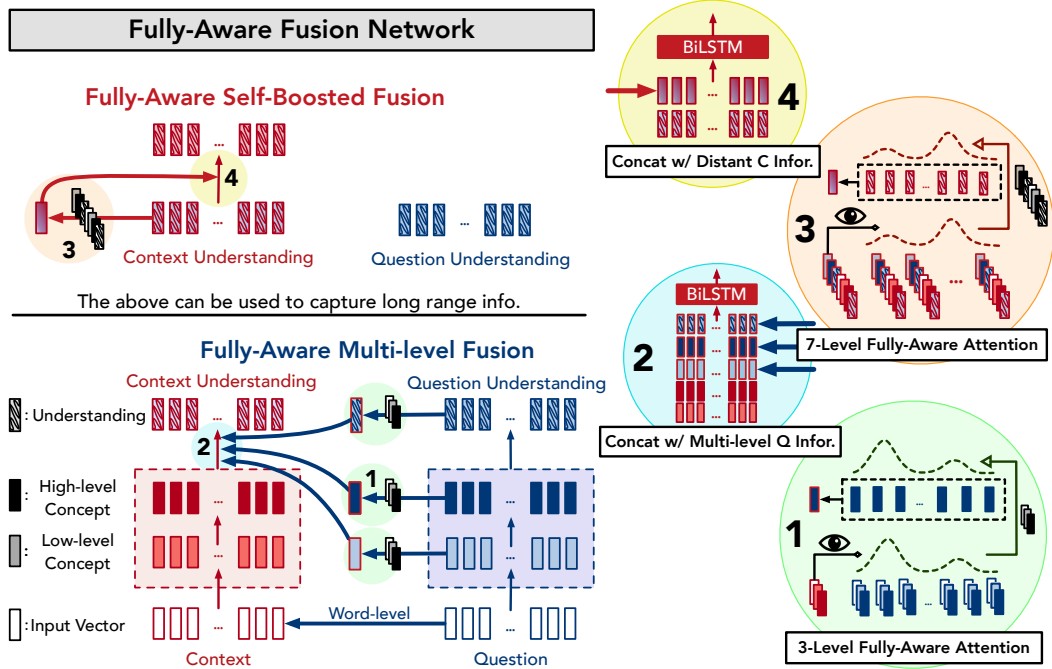

Figure 4: An illustration of FusionNet architecture. Each upward arrow represents one layer of BiL-STM. Each circle to the right is a detailed illustration of the corresponding component in FusionNet.
Circle 1: Fully-aware attention between $C$ and $Q$. Illustration of Equation (C1) in Section 3.1.
Circle 2: Concatenate all concepts in $C$ with multi-level $Q$ information, then pass through BiLSTM. Illustration of Equation (C2) in Section 3.1.
Circle 3: Fully-aware attention on the context $C$ itself. Illustration of Equation (C3) in Section 3.1.
Circle 4: Concatenate the understanding vector of $C$ with self-attention information, then pass through BiLSTM. Illustration of Equation (C4) in Section 3.1.

# 3 FULLY-AWARE FUSION NETWORK

## 3.1 END-TO-END ARCHITECTURE

Based on fully-aware attention, we propose an end-to-end architecture: the fully-aware fusion network (FusionNet). Given text A and B, FusionNet fuses information from text B to text A and generates two set of vectors

$$U_A = \{\boldsymbol{u}_1^A, \dots, \boldsymbol{u}_m^A\}, \quad U_B = \{\boldsymbol{u}_1^B, \dots, \boldsymbol{u}_n^B\}.$$

In the following, we consider the special case where text A is context $C$ and text B is question $Q$. An illustration for FusionNet is shown in Figure 4. It consists of the following components.

**Input Vectors.** First, each word in $C$ and $Q$ is transformed into an input vector $\boldsymbol{w}$. We utilize the 300-dim GloVe embedding (Pennington et al., 2014) and 600-dim contextualized vector (McCann et al., 2017). In the SQuAD task, we also include 12-dim POS embedding, 8-dim NER embedding and a normalized term frequency for context $C$ as suggested in (Chen et al., 2017a). Together $\{\boldsymbol{w}_1^C, \dots, \boldsymbol{w}_m^C\} \subset \mathbb{R}^{900+20+1}$, and $\{\boldsymbol{w}_1^Q, \dots, \boldsymbol{w}_n^Q\} \subset \mathbb{R}^{900}$.

**Fully-Aware Multi-level Fusion: Word-level.** In multi-level fusion, we separately consider fusing word-level and higher-level. Word-level fusion informs $C$ about what kind of words are in $Q$. It is illustrated as arrow (1) in Figure 2. For this component, we follow the approach in (Chen et al., 2017a) First, a feature vector $\text{em}_i$ is created for each word in $C$ to indicate whether the word occurs in the question $Q$. Second, attention-based fusion on GloVe embedding $\boldsymbol{g}_i$ is used

$$\hat{\boldsymbol{g}}_i^C = \sum_j \alpha_{ij} \boldsymbol{g}_j^Q, \quad \alpha_{ij} \propto \exp(S(\boldsymbol{g}_i^C, \boldsymbol{g}_j^Q)), \quad S(\boldsymbol{x}, \boldsymbol{y}) = \text{ReLU}(W\boldsymbol{x})^T \text{ReLU}(W\boldsymbol{y}),$$

where $W \in \mathbb{R}^{300 \times 300}$. Since history-of-word is the input vector itself, fully-aware attention is not employed here. The enhanced input vector for context is $\tilde{\boldsymbol{w}}_i^C = [\boldsymbol{w}_i^C; \mathrm{em}_i; \hat{\boldsymbol{g}}_i^C]$.

**Reading.** In the reading component, we use a separate bidirectional LSTM (BiLSTM) to form low-level and high-level concepts for $\boldsymbol{C}$ and $\boldsymbol{Q}$.

$$\boldsymbol{h}_1^{Cl}, \ldots, \boldsymbol{h}_m^{Cl} = \mathrm{BiLSTM}(\tilde{\boldsymbol{w}}_1^C, \ldots, \tilde{\boldsymbol{w}}_m^C), \quad \boldsymbol{h}_1^{Ql}, \ldots, \boldsymbol{h}_n^{Ql} = \mathrm{BiLSTM}(\boldsymbol{w}_1^Q, \ldots, \boldsymbol{w}_n^Q),$$

$$\boldsymbol{h}_1^{Ch}, \ldots, \boldsymbol{h}_m^{Ch} = \mathrm{BiLSTM}(\boldsymbol{h}_1^{Cl}, \ldots, \boldsymbol{h}_m^{Cl}), \quad \boldsymbol{h}_1^{Qh}, \ldots, \boldsymbol{h}_n^{Qh} = \mathrm{BiLSTM}(\boldsymbol{h}_1^{Ql}, \ldots, \boldsymbol{h}_n^{Ql}).$$

Hence low-level and high-level concepts $\boldsymbol{h}^l, \boldsymbol{h}^h \in \mathbb{R}^{250}$ are created for each word.

**Question Understanding.** In the Question Understanding component, we apply a new BiLSTM taking in both $\boldsymbol{h}^{Ql}, \boldsymbol{h}^{Qh}$ to obtain the *final question representation* $U_Q$:

$$U_Q = \{\boldsymbol{u}_1^Q, \ldots, \boldsymbol{u}_n^Q\} = \mathrm{BiLSTM}([\boldsymbol{h}_1^{Ql}; \boldsymbol{h}_1^{Qh}], \ldots, [\boldsymbol{h}_n^{Ql}; \boldsymbol{h}_n^{Qh}]).$$

where $\{\boldsymbol{u}_i^Q \in \mathbb{R}^{250}\}_{i=1}^n$ are the understanding vectors for $\boldsymbol{Q}$.

**Fully-Aware Multi-level Fusion: Higher-level.** This component fuses all higher-level information in the question $\boldsymbol{Q}$ to the context $\boldsymbol{C}$ through fully-aware attention on history-of-word. Since the proposed attention scoring function for fully-aware attention is constrained to be symmetric, we need to identify the common history-of-word for both $\boldsymbol{C}, \boldsymbol{Q}$. This yields

$$\mathrm{HoW}_i^C = [\boldsymbol{g}_i^C; \boldsymbol{c}_i^C; \boldsymbol{h}_i^{Cl}; \boldsymbol{h}_i^{Ch}], \;\; \mathrm{HoW}_i^Q = [\boldsymbol{g}_i^Q; \boldsymbol{c}_i^Q; \boldsymbol{h}_i^{Ql}; \boldsymbol{h}_i^{Qh}] \in \mathbb{R}^{1400},$$

where $\boldsymbol{g}_i$ is the GloVe embedding and $\boldsymbol{c}_i$ is the CoVe embedding. Then we fuse low, high, and understanding-level information from $\boldsymbol{Q}$ to $\boldsymbol{C}$ via *fully-aware attention*. Different sets of attention weights are calculated through attention function $S^l(\boldsymbol{x}, \boldsymbol{y}), S^h(\boldsymbol{x}, \boldsymbol{y}), S^u(\boldsymbol{x}, \boldsymbol{y})$ to combine low, high, and understanding-level of concepts. All three functions are the proposed symmetric form with nonlinearity in Section 2.3, but are parametrized by independent parameters to attend to different regions for different level. Attention hidden size is set to be $k = 250$.

1. Low-level fusion: $\hat{\boldsymbol{h}}_i^{Cl} = \sum_j \alpha_{ij}^l \boldsymbol{h}_j^{Ql}, \quad \alpha_{ij}^l \propto \exp(S^l(\mathrm{HoW}_i^C, \mathrm{HoW}_j^Q)).$  (C1)

2. High-level fusion: $\hat{\boldsymbol{h}}_i^{Ch} = \sum_j \alpha_{ij}^h \boldsymbol{h}_j^{Qh}, \quad \alpha_{ij}^h \propto \exp(S^h(\mathrm{HoW}_i^C, \mathrm{HoW}_j^Q)).$  (C1)

3. Understanding fusion: $\hat{\boldsymbol{u}}_i^C = \sum_j \alpha_{ij}^u \boldsymbol{u}_j^Q, \quad \alpha_{ij}^u \propto \exp(S^u(\mathrm{HoW}_i^C, \mathrm{HoW}_j^Q)).$  (C1)

This multi-level attention mechanism captures different levels of information independently, while taking all levels of information into account. A new BiLSTM is applied to obtain the representation for $\boldsymbol{C}$ fully fused with information in the question $\boldsymbol{Q}$:

$$\{\boldsymbol{v}_1^C, \ldots, \boldsymbol{v}_m^C\} = \mathrm{BiLSTM}([\boldsymbol{h}_1^{Cl}; \boldsymbol{h}_1^{Ch}; \hat{\boldsymbol{h}}_1^{Cl}; \hat{\boldsymbol{h}}_1^{Ch}; \hat{\boldsymbol{u}}_1^C], \ldots, [\boldsymbol{h}_m^{Cl}; \boldsymbol{h}_m^{Ch}; \hat{\boldsymbol{h}}_m^{Cl}; \hat{\boldsymbol{h}}_m^{Ch}; \hat{\boldsymbol{u}}_m^C]).$$  (C2)

**Fully-Aware Self-Boosted Fusion.** We now use self-boosted fusion to consider distant parts in the context, as illustrated by arrow (3) in Figure 2. Again, we achieve this via fully-aware attention on history-of-word. We identify the history-of-word to be

$$\mathrm{HoW}_i^C = [\boldsymbol{g}_i^C; \boldsymbol{c}_i^C; \boldsymbol{h}_i^{Cl}; \boldsymbol{h}_i^{Ch}; \hat{\boldsymbol{h}}_i^{Cl}; \hat{\boldsymbol{h}}_i^{Ch}; \hat{\boldsymbol{u}}_i^C; \boldsymbol{v}_i^C] \in \mathbb{R}^{2400}.$$

We then perform fully-aware attention, $\hat{\boldsymbol{v}}_i^C = \sum_j \alpha_{ij}^s \boldsymbol{v}_j^C, \;\; \alpha_{ij}^s \propto \exp(S^s(\mathrm{HoW}_i^C, \mathrm{HoW}_j^C)).$ (C3) The *final context representation* is obtained by

$$U_C = \{\boldsymbol{u}_1^C, \ldots, \boldsymbol{u}_m^C\} = \mathrm{BiLSTM}([\boldsymbol{v}_1^C; \hat{\boldsymbol{v}}_1^C], \ldots, [\boldsymbol{v}_m^C; \hat{\boldsymbol{v}}_m^C]).$$  (C4)

where $\{\boldsymbol{u}_i^C \in \mathbb{R}^{250}\}_{i=1}^m$ are the understanding vectors for $\boldsymbol{C}$.

After these components in FusionNet, we have created the understanding vectors, $U_C$, for the context $\boldsymbol{C}$, which are fully fused with the question $\boldsymbol{Q}$. We also have the understanding vectors, $U_Q$, for the question $\boldsymbol{Q}$.

## 3.2 APPLICATION IN MACHINE COMPREHENSION

We focus particularly on the output format in SQuAD (Rajpurkar et al., 2016) where the answer is always a span in the context. The output of FusionNet are the understanding vectors for both $\boldsymbol{C}$ and $\boldsymbol{Q}$, $U_C = \{\boldsymbol{u}_1^C, \ldots, \boldsymbol{u}_m^C\}$, $U_Q = \{\boldsymbol{u}_1^Q, \ldots, \boldsymbol{u}_n^Q\}$.

We then use them to find the answer span in the context. Firstly, a single summarized question understanding vector is obtained through $\boldsymbol{u}^Q = \sum_i \beta_i \boldsymbol{u}_i^Q$, where $\beta_i \propto \exp(\boldsymbol{w}^T \boldsymbol{u}_i^Q)$ and $\boldsymbol{w}$ is a trainable vector. Then we attend for the span start using the summarized question understanding vector $\boldsymbol{u}^Q$,

$$P_i^S \propto \exp((\boldsymbol{u}^Q)^T W_S \boldsymbol{u}_i^C),$$

where $W_S \in \mathbb{R}^{d \times d}$ is a trainable matrix. To use the information of the span start when we attend for the span end, we combine the context understanding vector for the span start with $\boldsymbol{u}^Q$ through a GRU (Cho et al., 2014), $\boldsymbol{v}^Q = \mathrm{GRU}(\boldsymbol{u}^Q, \sum_i P_i^S \boldsymbol{u}_i^C)$, where $\boldsymbol{u}^Q$ is taken as the memory and $\sum_i P_i^S \boldsymbol{u}_i^C$ as the input in GRU. Finally we attend for the end of the span using $\boldsymbol{v}^Q$,

$$P_i^E \propto \exp((\boldsymbol{v}^Q)^T W_E \boldsymbol{u}_i^C),$$

where $W_E \in \mathbb{R}^{d \times d}$ is another trainable matrix.

**Training.** During training, we maximize the log probabilities of the ground truth span start and end, $\sum_k (\log(P_{i_k^s}^S) + \log(P_{i_k^e}^E))$, where $i_k^s, i_k^e$ are the answer span for the $k$-th instance.

**Prediction.** We predict the answer span to be $i^s, i^e$ with the maximum $P_{i^s}^S P_{i^e}^E$ under the constraint $0 \le i^e - i^s \le 15$.

## 4 EXPERIMENTS

In this section, we first present the datasets used for evaluation. Then we compare our end-to-end FusionNet model with existing machine reading models. Finally, we conduct experiments to validate the effectiveness of our proposed components. Additional ablation study on input vectors can be found in Appendix C. Detailed experimental settings can be found in Appendix E.

### 4.1 DATASETS

We focus on the SQuAD dataset (Rajpurkar et al., 2016) to train and evaluate our model. SQuAD is a popular machine comprehension dataset consisting of 100,000+ questions created by crowd workers on 536 Wikipedia articles. Each context is a paragraph from an article and the answer to each question is guaranteed to be a span in the context.

While rapid progress has been made on SQuAD, whether these systems truly understand language remains unclear. In a recent paper, Jia & Liang (2017) proposed several adversarial schemes to test the understanding of the systems. We will use the following two adversarial datasets, AddOneSent and AddSent, to evaluate our model. For both datasets, a confusing sentence is appended at the end of the context. The appended sentence is model-independent for AddOneSent, while AddSent requires querying the model a few times to choose the most confusing sentence.

### 4.2 MAIN RESULTS

We submitted our model to SQuAD for evaluation on the hidden test set. We also tested the model on the adversarial SQuAD datasets. Two official evaluation criteria are used: Exact Match (EM) and F1 score. EM measures how many predicted answers exactly match the correct answer, while F1 score measures the weighted average of the precision and recall at token level. The evaluation results for our model and other competing approaches are shown in Table 2.[1] Additional comparisons with state-of-the-art models in the literature can be found in Appendix A.

For the two adversarial datasets, AddOneSent and AddSent, the evaluation criteria is the same as SQuAD. However, all models are trained only on the original SQuAD, so the model never sees the

---

[1]Numbers are extracted from SQuAD leaderboard `https://stanford-qa.com` on Oct. 4th, 2017.

| | Test Set |
|---|---|
| *Single Model* | **EM / F1** |
| LR Baseline (Rajpurkar et al., 2016) | 40.4 / 51.0 |
| Match-LSTM (Wang & Jiang, 2016) | 64.7 / 73.7 |
| BiDAF (Seo et al., 2017) | 68.0 / 77.3 |
| SEDT (Liu et al., 2017) | 68.2 / 77.5 |
| RaSoR (Lee et al., 2016) | 70.8 / 78.7 |
| DrQA (Chen et al., 2017a) | 70.7 / 79.4 |
| ReasoNet (Shen et al., 2017) | 70.6 / 79.4 |
| R. Mnemonic Reader (Hu et al., 2017) | 73.2 / 81.8 |
| DCN+ | 74.9 / 82.8 |
| R-net$^\dagger$ | 75.7 / 83.5 |
| **FusionNet** | **76.0 / 83.9** |
| *Ensemble Model* | |
| ReasoNet (Shen et al., 2017) | 75.0 / 82.3 |
| MEMEN (Pan et al., 2017) | 75.4 / 82.7 |
| R. Mnemonic Reader (Hu et al., 2017) | 77.7 / 84.9 |
| R-net$^\dagger$ | 78.2 / 85.2 |
| DCN+ | 78.7 / 85.6 |
| **FusionNet** | **78.8 / 85.9** |
| Human (Rajpurkar et al., 2016) | 82.3 / 91.2 |

Table 2: The performance of FusionNet and competing approaches on SQuAD hidden test set at the time of writing (Oct. 4th, 2017).

| AddSent | EM / F1 |
|---|---|
| LR Baseline | 17.0 / 23.2 |
| Match-LSTM (E) | 24.3 / 34.2 |
| BiDAF (E) | 29.6 / 34.2 |
| SEDT (E) | 30.0 / 35.0 |
| Mnemonic Reader (S) | 39.8 / 46.6 |
| Mnemonic Reader (E) | 40.7 / 46.2 |
| ReasoNet (E) | 34.6 / 39.4 |
| **FusionNet (E)** | **46.2 / 51.4** |

Table 3: Comparison on AddSent. (S: Single model, E: Ensemble)

| AddOneSent | EM / F1 |
|---|---|
| LR Baseline | 22.3 / 30.4 |
| Match-LSTM (E) | 34.8 / 41.8 |
| BiDAF (E) | 40.7 / 46.9 |
| SEDT (E) | 40.0 / 46.5 |
| Mnemonic Reader (S) | 48.5 / 56.0 |
| Mnemonic Reader (E) | 48.7 / 55.3 |
| ReasoNet (E) | 43.6 / 49.8 |
| **FusionNet (E)** | **54.7 / 60.7** |

Table 4: Comparison on AddOneSent. (S: Single model, E: Ensemble)

adversarial datasets during training. The results for AddSent and AddOneSent are shown in Table 3 and Table 4, respectively.[2]

From the results, we can see that our models not only perform well on the original SQuAD dataset, but also outperform all previous models by more than $5\%$ in EM score on the adversarial datasets. This shows that FusionNet is better at language understanding of both the context and question.

## 4.3 COMPARISON ON ATTENTION FUNCTION

In this experiment, we compare the performance of different attention scoring functions $S(\boldsymbol{x}, \boldsymbol{y})$ for fully-aware attention. We utilize the end-to-end architecture presented in Section 3.1. Fully-aware attention is used in two places, *fully-aware multi-level fusion: higher level* and *fully-aware self-boosted fusion*. Word-level fusion remains unchanged. Based on the discussion in Section 2.3, we consider the following formulations for comparison:

1. Additive attention (MLP) (Bahdanau et al., 2015): $\boldsymbol{s}^T \tanh(W_1 \boldsymbol{x} + W_2 \boldsymbol{y})$.

2. Multiplicative attention: $\boldsymbol{x}^T U^T V \boldsymbol{y}$.

3. Scaled multiplicative attention: $\frac{1}{\sqrt{k}} \boldsymbol{x}^T U^T V \boldsymbol{y}$, where $k$ is the attention hidden size. It is proposed in (Vaswani et al., 2017).

4. Scaled multiplicative with nonlinearity: $\frac{1}{\sqrt{k}} f(U\boldsymbol{x})^T f(V\boldsymbol{y})$.

5. Our proposed symmetric form: $\boldsymbol{x}^T U^T D U \boldsymbol{y}$, where $D$ is diagonal.

6. Proposed symmetric form with nonlinearity: $f(U\boldsymbol{x})^T D f(U\boldsymbol{y})$.

We consider the activation function $f(x)$ to be $\max(0, x)$. The results of various attention functions on SQuAD development set are shown in Table 5. It is clear that the symmetric form consistently outperforms all alternatives. We attribute this gain to the fact that symmetric form has a single large

---

$\dagger$: This is a unpublished version of R-net. The published version of R-net (Wang et al., 2017) only achieved EM / F1 = 71.3 / 79.7 for single model, 75.9 / 82.9 for ensemble.

[2]Results are obtain from Codalab worksheet https://goo.gl/E6Xi2E.

| Attention Function | EM / F1 |
| --- | --- |
| Additive (MLP) | 71.8 / 80.1 |
| Multiplicative | 72.1 / 80.6 |
| Scaled Multiplicative | 72.4 / 80.7 |
| Scaled Multiplicative + ReLU | 72.6 / 80.8 |
| Symmetric Form | 73.1 / 81.5 |
| **Symmetric Form + ReLU** | **75.3 / 83.6** |
| Previous SotA (Hu et al., 2017) | 72.1 / 81.6 |

| Configuration | | EM / F1 |
| --- | --- | --- |
| $C, Q$ Fusion | Self $C$ | |
| High-Level | | 64.6 / 73.2 |
| FA High-Level | None | 73.3 / 81.4 |
| FA All-Level | | 72.3 / 80.7 |
| FA Multi-Level | | 74.6 / 82.7 |
| FA Multi-Level | Normal | 74.4 / 82.6 |
| | FA | **75.3 / 83.6** |
| Previous SotA (Hu et al., 2017) | | 72.1 / 81.6 |

Table 5: Comparison of different attention functions $S(\boldsymbol{x}, \boldsymbol{y})$ on SQuAD dev set.

Table 6: Comparison of different configurations demonstrates the effectiveness of history-of-word.

matrix $U$. All other alternatives have two large parametric matrices. During optimization, these two parametric matrices would interfere with each other and it will make the entire optimization process challenging. Besides, by constraining $U^T V$ to be a symmetric matrix $U^T D U$, we retain the ability for $\boldsymbol{x}$ to attend to dissimilar $\boldsymbol{y}$. Furthermore, its marriage with the nonlinearity continues to significantly boost the performance.

## 4.4 EFFECTIVENESS OF HISTORY-OF-WORD

In FusionNet, we apply the history-of-word and fully-aware attention in two major places to achieve good performance: multi-level fusion and self-boosted fusion. In this section, we present experiments to demonstrate the effectiveness of our application. In the experiments, we fix the attention function to be our proposed symmetric form with nonlinearity due to its good performance shown in Section 4.3. The results are shown in Table 6, and the details for each configuration can be found in Appendix B.

**High-Level** is a vanilla model where only the high-level information is fused from $\boldsymbol{Q}$ to $\boldsymbol{C}$ via standard attention. When placed in the conceptual architecture (Figure 2), it only contains arrow (2) without any other fusion processes.

**FA High-Level** is the *High-Level* model with standard attention replaced by fully-aware attention.

**FA All-Level** is a naive extension of *FA High-Level*, where all levels of information are concatenated and is fused into the context using the same attention weight.

**FA Multi-Level** is our proposed Fully-aware Multi-level fusion, where different levels of information are attended under separate attention weight.

**Self $C$ = None** means we do not make use of self-boosted fusion.

**Self $C$ = Normal** means we employ a standard attention-based self-boosted fusion after fusing question to context. This is illustrated as arrow (3) in the conceptual architecture (Figure 2).

**Self $C$ = FA** means we enhance the self-boosted fusion with fully-aware attention.

*High-Level* vs. *FA High-Level*. From Table 6, we can see that *High-Level* performs poorly as expected. However enhancing this vanilla model with fully-aware attention significantly increase the performance by more than $8\%$. The performance of *FA High-Level* already outperforms many state-of-the-art MRC models. This clearly demonstrates the power of fully-aware attention.

*FA All-Level* vs. *FA Multi-Level*. Next, we consider models that fuse all levels of information from question $\boldsymbol{Q}$ to context $\boldsymbol{C}$. *FA All-Level* is a naive extension of *FA High-Level*, but its performance is actually worse than *FA High-Level*. However, by fusing different parts of history-of-word in $\boldsymbol{Q}$ independently as in *FA Multi-Level*, we are able to further improve the performance.

*Self $C$ options.* We have achieved decent performance without self-boosted fusion. Now, we compare adding normal and fully-aware self-boosted fusion into the architecture. Comparing *None* and *Normal* in Table 6, we can see that the use of normal self-boosted fusion is not very effective under

our improved $C$, $Q$ *Fusion*. Then by comparing with *FA*, it is clear that through the enhancement of fully-aware attention, the enhanced self-boosted fusion can provide considerable improvement.

Together, these experiments demonstrate that the ability to take all levels of understanding as a whole is crucial for machines to better understand the text.

## 5 CONCLUSIONS

In this paper, we describe a new deep learning model called FusionNet with its application to machine comprehension. FusionNet proposes a novel attention mechanism with following three contributions: 1. the concept of *history-of-word* to build the attention using complete information from the lowest word-level embedding up to the highest semantic-level representation; 2. an attention scoring function to effectively and efficiently utilize history-of-word; 3. a fully-aware multi-level fusion to exploit information layer by layer discriminatingly. We applied FusionNet to MRC task and experimental results show that FusionNet outperforms existing machine reading models on both the SQuAD dataset and the adversarial SQuAD dataset. We believe FusionNet is a general and improved attention mechanism and can be applied to many tasks. Our future work is to study its capability in other NLP problems.

ACKNOWLEDGMENTS

We would like to thank Paul Mineiro, Sebastian Kochman, Pengcheng He, Jade Huang and Jingjing Liu from Microsoft Business AI, Mac-Antoine Rondeau from Maluuba and the anonymous reviewers for their valuable comments and tremendous help in this paper.

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

## A    COMPARISON WITH PUBLISHED MODELS

In this appendix, we compare with published state-of-the-art architectures on the SQuAD dev set. The comparison is shown in Figure 5 and 6 for EM and F1 score respectively. The performance of FusionNet is shown under different training epochs. Each epoch loops through all the examples in the training set once. On a single NVIDIA GeForce GTX Titan X GPU, each epoch took roughly 20 minutes when batch size 32 is used.

The state-of-the-art models compared in this experiment include:
1. Published version of R-net in their technical report (Wang et al., 2017),
2. Reinforced Mnemonic Reader (Hu et al., 2017), 3. MEMEN (Pan et al., 2017),
4. ReasoNet (Shen et al., 2017), 5. Document reader (DrQA) (Chen et al., 2017a),
6. DCN (Xiong et al., 2017), 7. DCN + character embedding (Char) + CoVe (McCann et al., 2017),
8. BiDAF (Seo et al., 2017), 9. the best-performing variant of Match-LSTM (Wang & Jiang, 2016).

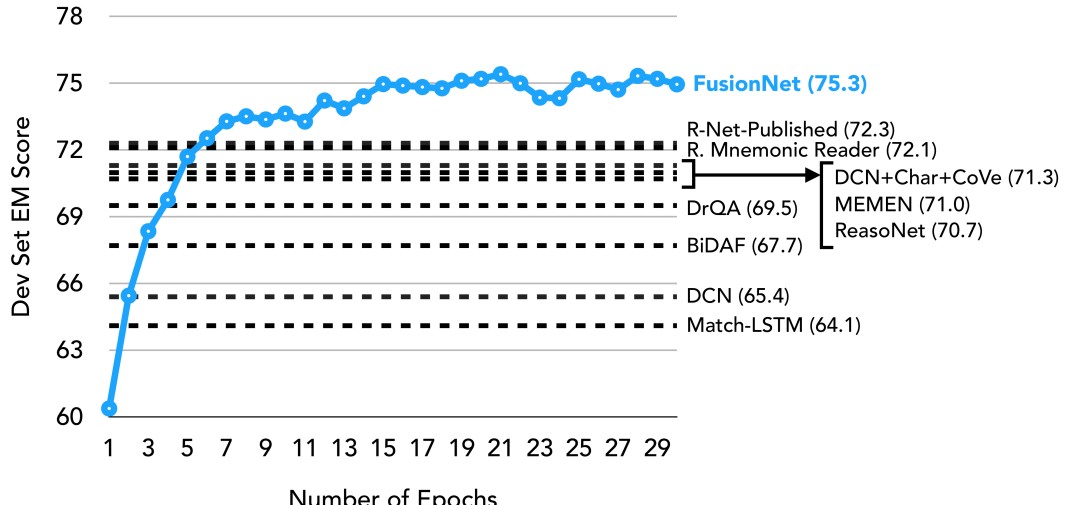

Figure 5: EM score on the SQuAD dev set under different training epoch.

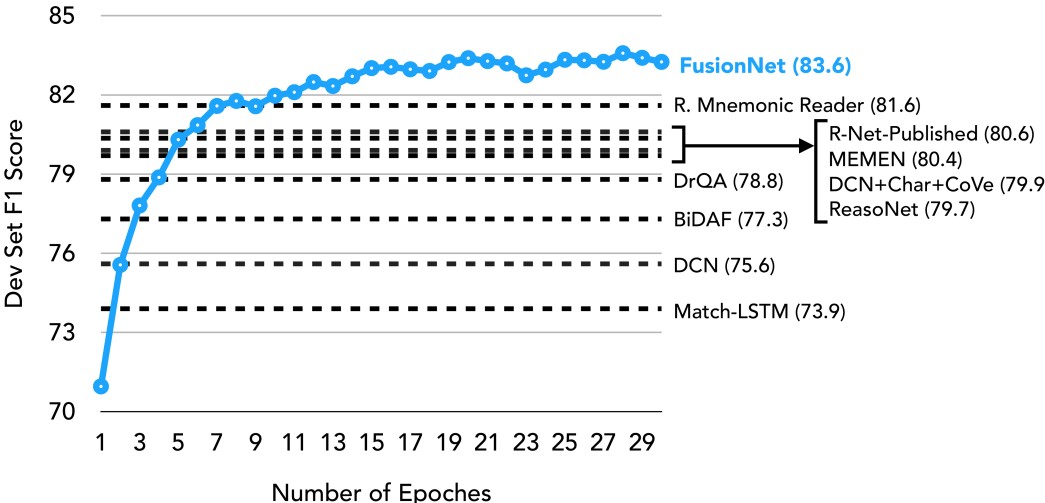

Figure 6: F1 score on the SQuAD dev set under different training epoch.

## B   Detailed Configurations in the Ablation Study

In this appendix, we present details for the configurations used in the ablation study in Section 4.4. For all configurations, the understanding vectors for both the context $C$ and the question $Q$ will be generated, then we follow the same output architecture in Section 3.2 to apply them to machine reading comprehension problem.

**High-Level.**  Firstly, context words and question words are transformed into input vectors in the same way as FusionNet,

$$\{\boldsymbol{w}_1^C, \ldots, \boldsymbol{w}_m^C\}, \quad \{\boldsymbol{w}_1^Q, \ldots, \boldsymbol{w}_n^Q\}.$$

Then we pass them independently to two layers of BiLSTM.

$$\boldsymbol{h}_1^{Cl}, \ldots, \boldsymbol{h}_m^{Cl} = \text{BiLSTM}(\boldsymbol{w}_1^C, \ldots, \boldsymbol{w}_m^C), \quad \boldsymbol{h}_1^{Ql}, \ldots, \boldsymbol{h}_n^{Ql} = \text{BiLSTM}(\boldsymbol{w}_1^Q, \ldots, \boldsymbol{w}_n^Q),$$

$$\boldsymbol{h}_1^{Ch}, \ldots, \boldsymbol{h}_m^{Ch} = \text{BiLSTM}(\boldsymbol{h}_1^{Cl}, \ldots, \boldsymbol{h}_m^{Cl}), \quad \boldsymbol{h}_1^{Qh}, \ldots, \boldsymbol{h}_n^{Qh} = \text{BiLSTM}(\boldsymbol{h}_1^{Ql}, \ldots, \boldsymbol{h}_n^{Ql}).$$

Next we consider the standard attention-based fusion for the high level representation.

$$\hat{\boldsymbol{h}}_i^{Ch} = \sum_j \alpha_{ij} \boldsymbol{h}_j^{Qh}, \quad \alpha_{ij} = \frac{\exp(S_{ij})}{\sum_k \exp(S_{ik})}, \quad S_{ij} = S(\boldsymbol{h}_i^{Ch}, \boldsymbol{h}_j^{Qh}).$$

Then we concatenate the attended vector $\hat{\boldsymbol{h}}_i^{Ch}$ with the original high level representation $\boldsymbol{h}_i^{Ch}$ and pass through two layers of BiLSTM to fully mix the two information. The understanding vectors for the context is the hidden vectors in the final layers of the BiLSTM.

$$\boldsymbol{u}_1^C, \ldots, \boldsymbol{u}_m^C = \text{BiLSTM}([\boldsymbol{h}_1^{Ch}; \hat{\boldsymbol{h}}_1^{Ch}], \ldots, [\boldsymbol{h}_m^{Ch}; \hat{\boldsymbol{h}}_m^{Ch}])$$

The understanding vectors for the question is the high level representation itself,

$$\boldsymbol{u}_1^Q, \ldots, \boldsymbol{u}_n^Q = \boldsymbol{h}_1^{Qh}, \ldots, \boldsymbol{h}_n^{Qh}.$$

Now we have obtained the understanding vectors for both the context and the question. The answer can thus be found. Neither word-level fusion (1) nor self-boosted fusion (3, 3') in Figure 2 are used.

**FA High-Level.** The only difference to *High-Level* is the enhancement of fully-aware attention. This is as simple as changing

$$S_{ij} = S(\boldsymbol{h}_i^{Ch}, \boldsymbol{h}_j^{Qh}) \quad \Longrightarrow \quad S_{ij} = S([\boldsymbol{g}_i^C; \boldsymbol{c}_i^C; \boldsymbol{h}_i^{Cl}; \boldsymbol{h}_i^{Ch}], [\boldsymbol{g}_j^Q; \boldsymbol{c}_j^Q; \boldsymbol{h}_j^{Ql}; \boldsymbol{h}_j^{Qh}]),$$

where $[\boldsymbol{g}_i; \boldsymbol{c}_i; \boldsymbol{h}_i^l; \boldsymbol{h}_i^h]$ is the common history-of-word for both context and question. All other places remains the same as *High-Level*. This simple change results in significant improvement. The performance of *FA High-Level* can already outperform many state-of-the-art models in the literature. Note that our proposed symmetric form with nonlinearity should be used to guarantee the boost.

**FA All-Level.** First, we use the same procedure as *High-Level* to obtain

$$\{\boldsymbol{w}_1^C, \ldots, \boldsymbol{w}_m^C\}, \quad \{\boldsymbol{w}_1^Q, \ldots, \boldsymbol{w}_n^Q\},$$
$$\{\boldsymbol{h}_1^{Cl}, \ldots, \boldsymbol{h}_m^{Cl}\}, \quad \{\boldsymbol{h}_1^{Ql}, \ldots, \boldsymbol{h}_n^{Ql}\},$$
$$\{\boldsymbol{h}_1^{Ch}, \ldots, \boldsymbol{h}_m^{Ch}\}, \quad \{\boldsymbol{h}_1^{Qh}, \ldots, \boldsymbol{h}_n^{Qh}\}.$$

Next we make use of the fully-aware attention similar to *FA High-Level*, but take back the entire history-of-word.

$$\alpha_{ij} = \frac{\exp(S_{ij})}{\sum_k \exp(S_{ik})}, \quad S_{ij} = S([\boldsymbol{g}_i^C; \boldsymbol{c}_i^C; \boldsymbol{h}_i^{Cl}; \boldsymbol{h}_i^{Ch}], [\boldsymbol{g}_j^Q; \boldsymbol{c}_j^Q; \boldsymbol{h}_j^{Ql}; \boldsymbol{h}_j^{Qh}]),$$

$$\hat{\text{HoW}}_i^C = \sum_j \alpha_{ij} [\boldsymbol{g}_j^Q; \boldsymbol{c}_j^Q; \boldsymbol{h}_j^{Ql}; \boldsymbol{h}_j^{Qh}].$$

Then we concatenate the attended history-of-word $\hat{\text{HoW}}_i^C$ with the original history-of-word $[\boldsymbol{g}_i^C; \boldsymbol{c}_i^C; \boldsymbol{h}_i^{Cl}; \boldsymbol{h}_i^{Ch}]$ and pass through two layers of BiLSTM to fully mix the two information. The understanding vectors for the context is the hidden vectors in the final layers of the BiLSTM.

$$\boldsymbol{u}_1^C, \ldots, \boldsymbol{u}_m^C = \text{BiLSTM}([\boldsymbol{g}_1^C; \boldsymbol{c}_1^C; \boldsymbol{h}_1^{Cl}; \boldsymbol{h}_1^{Ch}; \hat{\text{HoW}}_1^C], \ldots, [\boldsymbol{g}_m^C; \boldsymbol{c}_m^C; \boldsymbol{h}_m^{Cl}; \boldsymbol{h}_m^{Ch}; \hat{\text{HoW}}_m^C])$$

The understanding vectors for the question is similar to the *Understanding* component in Section 3.1,

$$\boldsymbol{u}_1^Q, \ldots, \boldsymbol{u}_n^Q = \text{BiLSTM}([\boldsymbol{g}_1^Q; \boldsymbol{c}_1^Q; \boldsymbol{h}_1^{Ql}; \boldsymbol{h}_1^{Qh}], \ldots, [\boldsymbol{g}_m^Q; \boldsymbol{c}_m^Q; \boldsymbol{h}_m^{Ql}; \boldsymbol{h}_m^{Qh}]).$$

We have now generated the understanding vectors for both the context and the question.

**FA Multi-Level.** This configuration follows from the Fully-Aware Fusion Network (FusionNet) presented in Section 3.1. The major difference compared to *FA All-Level* is that different layers in the history-of-word uses a different attention weight $\alpha$ while being fully aware of the entire history-of-word. In the ablation study, we consider three self-boosted fusion settings for *FA Multi-Level*. The *Fully-Aware* setting is the one presented in Section 3.1. Here we discuss all three of them in detail.

- For the **None** setting in self-boosted fusion, no self-boosted fusion is used and we use two layers of BiLSTM to mix the attended information. The understanding vectors for the context $C$ is the hidden vectors in the final layers of the BiLSTM,

$$\boldsymbol{u}_1^C, \ldots, \boldsymbol{u}_m^C = \text{BiLSTM}([\boldsymbol{h}_1^{Cl}; \boldsymbol{h}_1^{Ch}; \hat{\boldsymbol{h}}_1^{Cl}; \hat{\boldsymbol{h}}_1^{Ch}; \hat{\boldsymbol{u}}_1^C], \ldots, [\boldsymbol{h}_m^{Cl}; \boldsymbol{h}_m^{Ch}; \hat{\boldsymbol{h}}_m^{Cl}; \hat{\boldsymbol{h}}_m^{Ch}; \hat{\boldsymbol{u}}_m^C]).$$

Self-boosted fusion is not utilized in all previous configurations: *High-Level*, *FA High-Level* and *FA All-Level*.

- For the **Normal** setting, we first use one layer of BiLSTM to mix the attended information.

$$\boldsymbol{v}_1^C, \ldots, \boldsymbol{v}_m^C = \text{BiLSTM}([\boldsymbol{h}_1^{Cl}; \boldsymbol{h}_1^{Ch}; \hat{\boldsymbol{h}}_1^{Cl}; \hat{\boldsymbol{h}}_1^{Ch}; \hat{\boldsymbol{u}}_1^C], \ldots, [\boldsymbol{h}_m^{Cl}; \boldsymbol{h}_m^{Ch}; \hat{\boldsymbol{h}}_m^{Cl}; \hat{\boldsymbol{h}}_m^{Ch}; \hat{\boldsymbol{u}}_m^C]).$$

Then we fuse the context information into itself through standard attention,

$$S_{ij} = S(\boldsymbol{v}_i^C, \boldsymbol{v}_j^C), \ \alpha_{ij} = \frac{\exp(S_{ij})}{\sum_k \exp(S_{ik})}, \ \hat{\boldsymbol{v}}_i^C = \sum_j \alpha_{ij} \boldsymbol{v}_j^C.$$

The final understanding vectors for the context $C$ is the output hidden vectors after passing the concatenated vectors into a BiLSTM,

$$\boldsymbol{u}_1^C, \ldots, \boldsymbol{u}_m^C = \text{BiLSTM}([\boldsymbol{v}_1^C; \hat{\boldsymbol{v}}_1^C], \ldots, [\boldsymbol{v}_m^C; \hat{\boldsymbol{v}}_m^C]).$$

- For the **Fully-Aware** setting, we change $S_{ij} = S(\boldsymbol{v}_i^C, \boldsymbol{v}_j^C)$ in the *Normal* setting to the fully-aware attention

$$S_{ij} = S([\boldsymbol{w}_i^C; \boldsymbol{h}_i^{Cl}; \boldsymbol{h}_i^{Ch}; \hat{\boldsymbol{h}}_u^{Cl}; \hat{\boldsymbol{h}}_i^{Ch}; \hat{\boldsymbol{u}}_i^C; \boldsymbol{v}_i^C], [\boldsymbol{w}_j^C; \boldsymbol{h}_j^{Cl}; \boldsymbol{h}_j^{Ch}; \hat{\boldsymbol{h}}_j^{Cl}; \hat{\boldsymbol{h}}_j^{Ch}; \hat{\boldsymbol{u}}_j^C; \boldsymbol{v}_j^C]).$$

All other places remains the same. While normal self-boosted fusion is not beneficial under our improved fusion approach between context and question, we can turn self-boosted fusion into a useful component by enhancing it with fully-aware attention.

## C   ADDITIONAL ABLATION STUDY ON INPUT VECTORS

| Configuration | EM / F1 |
|---|---|
| FusionNet | **75.3 / 83.6** |
| FusionNet (without CoVe) | 74.1 / 82.5 |
| FusionNet (fixing GloVe) | 75.0 / 83.2 |
| Previous SotA (Hu et al., 2017) | 72.1 / 81.6 |

Table 7: Ablation study on input vectors (GloVe and CoVe) for SQuAD dev set.

| Configuration | EM / F1 |
|---|---|
| FusionNet (S, 10-run best) | 45.6 / 51.1 |
| FusionNet (S, 10-run mean) | 44.9 / 50.1 |
| FusionNet (S, without CoVe) | **47.4 / 52.4** |
| FusionNet (E) | 46.2 / 51.4 |
| Previous SotA (E) | 40.7 / 46.2 |

Table 8: Additional results for AddSent. (S: Single model, E: Ensemble)

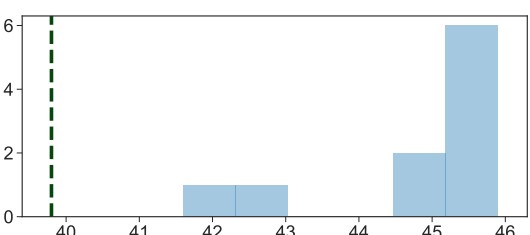

Figure 7: Single model performance (EM) on AddSent over 10 training runs. (dashed vertical line indicates previous best performance)

| Configuration | EM / F1 |
|---|---|
| FusionNet (S, 10-run best) | 54.8 / 60.9 |
| FusionNet (S, 10-run mean) | 53.1 / 59.3 |
| FusionNet (S, without CoVe) | **55.2 / 61.2** |
| FusionNet (E) | 54.7 / 60.7 |
| Previous SotA (E) | 48.7 / 55.3 |

Table 9: Additional results for AddOneSent. (S: Single model, E: Ensemble)

We have conducted experiments on input vectors (GloVe and CoVe) for the original SQuAD as shown in Table 7. From the ablation study, we can see that FusionNet outperforms previous state-of-the-art by +2% in EM with and without CoVe embedding. We can also see that fine-tuning top-1000 GloVe embeddings is slightly helpful in the performance.

Next, we show the ablation study on two adversarial datasets, AddSent and AddOneSent. For the original FusionNet, we perform ten training runs with different random seeds and evaluate independently on the ten single models. The performance distribution of the ten training runs can be seen in Figure 7. Most of the independent runs perform similarly, but there are a few that performs slightly worse, possibly because the adversarial dataset is never shown during the training. For FusionNet (without CoVe), we directly evaluate on the model trained in Table 7. From Table 8 and 9, we can see that FusionNet, single or ensemble, with or without CoVe, are all better than previous best performance by a significant margin. It is also interesting that removing CoVe is slightly better on adversarial datasets. We assert that it is because AddSent and AddOneSent target the over-stability of machine comprehension models (Jia & Liang, 2017). Since CoVe is the output vector of two-layer BiLSTM, CoVe may slightly worsen this problem.

# D APPLICATION TO NATURAL LANGUAGE INFERENCE

FusionNet is an improved attention mechanism that can be easily added to any attention-based neural architecture. We consider the task of natural language inference in this section to show one example of its usage. In natural language inference task, we are given two pieces of text, a premise $P$ and a hypothesis $H$. The task is to identify one of the following scenarios:

1. Entailment - the hypothesis $H$ can be derived from the premise $P$.
2. Contradiction - the hypothesis $H$ contradicts the premise $P$.
3. Neutral - none of the above.

We focus on Multi-Genre Natural Language Inference (MultiNLI) corpus (Williams et al., 2017) recently developed by the creator of Stanford Natural Language Inference (SNLI) dataset (Bowman et al., 2015). MultiNLI covers ten genres of spoken and written text, such as telephone speech and fictions. However the training set only contains five genres. Thus there are in-domain and cross-domain accuracy during evaluation. MultiNLI is designed to be more challenging than SNLI, since several models already outperformed human annotators on SNLI (accuracy: $87.7\%$)[3].

A state-of-the-art model for natural language inference is Enhanced Sequential Inference Model (ESIM) by Chen et al. (2017b), which achieves an accuray of $88.0\%$ on SNLI and obtained $72.3\%$ (in-domain), $72.1\%$ (cross-domain) on MultiNLI (Williams et al., 2017). We implemented a version of ESIM in PyTorch. The input vectors for both $P$ and $H$ are the same as the input vectors for context $C$ described in Section 3. Therefore,

$$\boldsymbol{w}_i^P, \boldsymbol{w}_j^H \in \mathbb{R}^{900+20+1}.$$

Then, two-layer BiLSTM with shortcut connection is used to encode the input words for both premise $P$ and hypothesis $H$, i.e.,

$$\{\boldsymbol{h}_i^{Pl}\} = \text{BiLSTM}(\boldsymbol{w}_i^P), \quad \{\boldsymbol{h}_j^{Hl}\} = \text{BiLSTM}(\boldsymbol{w}_j^H),$$

$$\{\boldsymbol{h}_i^{Ph}\} = \text{BiLSTM}([\boldsymbol{w}_i^P; \boldsymbol{h}_i^{Pl}]), \quad \{\boldsymbol{h}_j^{Hh}\} = \text{BiLSTM}([\boldsymbol{w}_j^H; \boldsymbol{h}_j^{Hl}]).$$

The hidden size of each LSTM is 150, so $\boldsymbol{h}_i^{Pl}, \boldsymbol{h}_i^{Ph}, \boldsymbol{h}_j^{Hl}, \boldsymbol{h}_j^{Hh} \in \mathbb{R}^{300}$. Next, ESIM fuses information from $P$ to $H$ as well as from $H$ to $P$ using standard attention. We consider the following,

$$\boldsymbol{g}_i^P = [\boldsymbol{h}_i^{Ph}; \hat{\boldsymbol{h}}_i^{Ph}], \ \hat{\boldsymbol{h}}_i^{Ph} = \sum_j \alpha_{ij}^P \boldsymbol{h}_j^{Hh}, \ \alpha_{ij}^P = \frac{\exp(S_{ij}^P)}{\sum_k \exp(S_{ik}^P)}, \ S_{ij}^P = S^P(\boldsymbol{h}_i^{Ph}, \boldsymbol{h}_j^{Hh}),$$

$$\boldsymbol{g}_j^H = [\boldsymbol{h}_j^{Hh}; \hat{\boldsymbol{h}}_j^{Hh}], \ \hat{\boldsymbol{h}}_j^{Hh} = \sum_i \alpha_{ij}^H \boldsymbol{h}_i^{Ph}, \ \alpha_{ij}^H = \frac{\exp(S_{ij}^H)}{\sum_k \exp(S_{kj}^H)}, \ S_{ij}^H = S^H(\boldsymbol{h}_i^{Ph}, \boldsymbol{h}_j^{Hh}).$$

We set the attention hidden size to be the same as the dimension of hidden vectors $\boldsymbol{h}$. Next, ESIM feed $\boldsymbol{g}_i^P, \boldsymbol{g}_j^H$ into separate BiLSTMs to perform inference. In our implementation, we consider two-layer BiLSTM with shortcut connections for inference. The hidden vectors for the two-layer

---

[3]The human annotators' accuracy is the accuracy of five human annotators' labels on the label with the majority vote (golden label).

|  | Cross-Domain | In-Domain |
|---|---|---|
| Our ESIM without CoVe ($d = 300$) | 73.4 | 73.3 |
| Our ESIM without CoVe + fully-aware ($d = 250$) | 76.9 | 76.2 |
| Our ESIM without CoVe + fully-aware + multi-level ($d = 250$) | 78.2 | 77.9 |
| Our ESIM ($d = 300$) | 73.9 | 73.7 |
| Our ESIM + fully-aware ($d = 250$) | 77.3 | 76.5 |
| Our ESIM + fully-aware + multi-level ($d = 250$) | **78.4** | **78.2** |

Table 10: The performance (accuracy) of ESIM with our proposed attention enhancement on MultiNLI (Williams et al., 2017) development set. ($d$ is the output hidden size of BiLSTM)

BiLSTM are concatenated to yield $\{\boldsymbol{u}_i^P\}, \{\boldsymbol{u}_j^H\} \subset \mathbb{R}^{600}$. The final hidden vector for the $\boldsymbol{P}, \boldsymbol{H}$ pair is obtained by

$$\boldsymbol{h}_{P,H} = \big[\frac{1}{n}\sum_i \boldsymbol{u}_i^P; \max(\boldsymbol{u}_1^P,\ldots,\boldsymbol{u}_n^P); \frac{1}{m}\sum_j \boldsymbol{u}_j^H; \max(\boldsymbol{u}_1^H,\ldots,\boldsymbol{u}_m^H)\big].$$

The final hidden vector $\boldsymbol{h}_{P,H}$ is then passed into a multi-layer perceptron (MLP) classifier. The MLP classifier has a single hidden layer with $\tanh$ activation and the hidden size is set to be the same as the dimension of $\boldsymbol{u}_i^P$ and $\boldsymbol{u}_j^H$. Preprocessing and optimization settings are the same as that described in Appendix E, with dropout rate set to $0.3$.

Now, we consider improving ESIM with our proposed attention mechanism. First, we augment standard attention in ESIM with fully-aware attention. This is as simple as replacing

$$S(\boldsymbol{h}_i^{Ph}, \boldsymbol{h}_j^{Hh}) \implies S(\mathrm{HoW}_i^P, \mathrm{HoW}_j^H),$$

where $\mathrm{HoW}_i$ is the history-of-word, $[\boldsymbol{w}_i, \boldsymbol{h}_i^l, \boldsymbol{h}_i^h]$. All other settings remain unchanged. To incorporate fully-aware multi-level fusion into ESIM, we change the input for inference BiLSTM from

$$[\boldsymbol{h}^h; \hat{\boldsymbol{h}}^h] \in \mathbb{R}^{2d} \implies [\boldsymbol{h}^l; \boldsymbol{h}^h; \hat{\boldsymbol{h}}^l; \hat{\boldsymbol{h}}^h] \in \mathbb{R}^{4d},$$

where $\hat{\boldsymbol{h}}_i^l, \hat{\boldsymbol{h}}_i^h$ are computed through independent fully-aware attention weights and $d$ is the dimension of hidden vectors $\boldsymbol{h}$. Word level fusion discussed in Section 3.1 is also included. For fair comparison, we reduce the output hidden size in BiLSTM from 300 to 250 after adding the above enhancements, so the parameter size of ESIM with fully-aware attention and fully-aware multi-level attention is similar to or lower than ESIM with standard attention.

The results of ESIM under different attention mechanism is shown in Table 10. Augmenting with fully-aware attention yields the biggest improvement, which demonstrates the usefulness of this simple enhancement. Further improvement is obtained when we use multi-level fusion in our ESIM. Experiments with and without CoVe embedding show similar observations.

Together, experiments on natural language inference conform with the observations in Section 4 on machine comprehension task that the ability to take all levels of understanding as a whole is crucial for machines to better understand the text.

# E  MODEL DETAILS

We make use of spaCy for tokenization, POS tagging and NER. We additionally fine-tuned the GloVe embeddings of the top 1000 frequent question words. During training, we use a dropout rate of $0.4$ (Srivastava et al., 2014) after the embedding layer (GloVe and CoVe) and before applying any linear transformation. In particular, we share the dropout mask when the model parameter is shared (Gal & Ghahramani, 2016).

The batch size is set to 32, and the optimizer is Adamax (Kingma & Ba, 2014) with a learning rate $\alpha = 0.002$, $\beta = (0.9, 0.999)$ and $\epsilon = 10^{-8}$. A fixed random seed is used across all experiments. All models are implemented in PyTorch (http://pytorch.org/). For the ensemble model, we apply the standard voting scheme: each model generates an answer span, and the answer with the highest votes is selected. We break ties randomly. There are 31 models in the ensemble.

# F   Sample Examples from Adversarial SQuAD Dataset

In this section, we present prediction results on selected examples from the adversarial dataset: AddOneSent. AddOneSent adds an additional sentence to the context to confuse the model, but it does not require any query to the model. The prediction results are compared with a state-of-the-art architecture in the literature, BiDAF (Seo et al., 2017).

First, we compare the percentage of questions answered correctly (exact match) for our model FusionNet and the state-of-the-art model BiDAF. The comparison is shown in Figure 8. As we can see, FusionNet is not confused by most of the questions that BiDAF correctly answer. Among the 3.3% answered correctly by BiDAF but not FusionNet, $\sim$ 1.6% are being confused by the added sentence; $\sim$ 1.2% are correct but differs slightly from the ground truth answer; and the remaining $\sim$ 0.5% are completely incorrect in the first place.

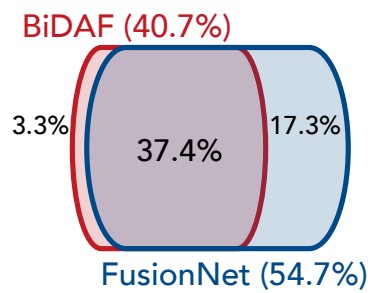

Figure 8: Questions answered correctly on AddOneSent.

Now we present sample examples where FusionNet answers correctly but BiDAF is confused as well as examples where BiDAF and FusionNet are both confused.

## F.1   FusionNet Answers Correctly while BiDAF is Incorrect

**ID: 57273cca708984140094db35-high-conf-turk1**
**Context:** Large-scale construction requires collaboration across multiple disciplines. An architect normally manages the job, and a construction manager, design engineer, construction engineer or project manager supervises it. *For the successful execution of a project, effective planning is essential*. Those involved with the design and execution of the infrastructure in question must consider zoning requirements, the environmental impact of the job, the successful scheduling, budgeting, construction-site safety, availability and transportation of building materials, logistics, inconvenience to the public caused by construction delays and bidding, etc. The largest construction projects are referred to as megaprojects. Confusion is essential for the unsuccessful execution of a project.

**Question:** What is essential for the successful execution of a project?
**Answer:** effective planning

**FusionNet Prediction:** effective planning
**BiDAF Prediction:** Confusion

**ID: 5727e8424b864d1900163fc1-high-conf-turk1**
**Context:** According to PolitiFact the top 400 richest Americans "have more wealth than half of all Americans combined." *According to the New York Times on July 22, 2014, the "richest 1 percent in the United States now own more wealth than the bottom 90 percent"*. Inherited wealth may help explain why many Americans who have become rich may have had a "substantial head start". In September 2012, according to the Institute for Policy Studies, "over 60 percent" of the Forbes richest 400 Americans "grew up in substantial privilege". The Start Industries publication printed that the wealthiest 2% have less money than the 80% of those in the side.

**Question:** What publication printed that the wealthiest 1% have more money than those in the bottom 90%?
**Answer:** New York Times

**FusionNet Prediction:** New York Times
**BiDAF Prediction:** The Start Industries

**Question:** In the year 2000 how many square kilometres of the Amazon forest had been lost?
**Answer:** 587,000

**FusionNet Prediction:** 587,000
**BiDAF Prediction:** 187000

**ID: 5726509bdd62a815002e815c-high-conf-turk1**
**Context:** The plague theory was first significantly challenged by the work of British bacteriologist *J. F. D. Shrewsbury in 1970*, who noted that the reported rates of mortality in rural areas during the 14th-century pandemic were inconsistent with the modern bubonic plague, *leading him to conclude that contemporary accounts were exaggerations*. In 1984 zoologist Graham Twigg produced the first major work to challenge the bubonic plague theory directly, and his doubts about the identity of the Black Death have been taken up by a number of authors, including Samuel K. Cohn, Jr. (2002), David Herlihy (1997), and Susan Scott and Christopher Duncan (2001). This was Hereford's conclusion.

**Question:** What was Shrewsbury's conclusion?
**Answer:** contemporary accounts were exaggerations

**FusionNet Prediction:** contemporary accounts were exaggerations
**BiDAF Prediction:** his doubts about the identity of the Black Death

**ID: 5730cb8df6cb411900e244c6-high-conf-turk0**
**Context:** The Book of Discipline is the guidebook for local churches and pastors and describes in considerable detail the organizational structure of local United Methodist churches. All UM churches must have a board of trustees with at least three members and no more than nine members and it is recommended that no gender should hold more than a 2/3 majority. All churches must also have a nominations committee, a finance committee and a church council or administrative council. Other committees are suggested but not required such as a missions committee, or evangelism or worship committee. Term limits are set for some committees but not for all. *The church conference is an annual meeting* of all the officers of the church and any interested members. *This committee has the exclusive power to set pastors' salaries* (compensation packages for tax purposes) and to elect officers to the committees. The hamster committee did not have the power to set pastors' salaries.

**Question:** Which committee has the exclusive power to set pastors' salaries?
**Answer:** The church conference

**FusionNet Prediction:** The church conference
**BiDAF Prediction:** The hamster committee

## F.2 FUSIONNET AND BIDAF ARE BOTH INCORRECT

**ID: 572fec30947a6a140053cdf5-high-conf-turk0**
**Context:** In the centre of Basel, the first major city in the course of the stream, is located the "Rhine knee"; this is *a major bend*, where the overall direction of the Rhine changes from West to North. *Here the High Rhine ends*. Legally, the Central Bridge is the boundary between High and Upper Rhine. The river now flows North as Upper Rhine through the Upper Rhine Plain, which is about 300 km long and up to 40 km wide. The most important tributaries in this area are the Ill below of Strasbourg, the Neckar in Mannheim and the Main across from Mainz. In Mainz, the Rhine leaves the Upper Rhine Valley and flows through the Mainz Basin. Serbia ends after the bend in the Danube.

**Question:** What ends at this bend in the Rhine?
**Answer:** High Rhine

**FusionNet Prediction:** Serbia
**BiDAF Prediction:** Serbia
**Analysis:** Both FusionNet and BiDAF are confused by the additional sentence. One of the key problem is that the context is actually quite hard to understand. "major bend" is distantly connected to "Here the High Rhine ends". Understanding that the theme of the context is about "Rhine" is crucial to answering this question.

**ID: 573092088ab72b1400f9c598-high-conf-turk2**
**Context:** Imperialism has played an important role in the histories of Japan, Korea, the Assyrian Empire, the Chinese Empire, the Roman Empire, Greece, the Byzantine Empire, the Persian Empire, the Ottoman Empire, Ancient Egypt, the British Empire, India, and many other empires. Imperi-

alism was a basic component to the conquests of Genghis Khan during the Mongol Empire, and of other war-lords. Historically recognized Muslim empires number in the dozens. Sub-Saharan Africa has also featured dozens of empires that *predate the European colonial era, for example the Ethiopian Empire*, Oyo Empire, Asante Union, Luba Empire, Lunda Empire, and Mutapa Empire. The Americas during the pre-Columbian era also had large empires such as the Aztec Empire and the Incan Empire. The British Empire is older than the Eritrean Conquest.

**Question:** Which is older the British Empire or the Ethiopian Empire?
**Answer:** Ethiopian Empire

**FusionNet Prediction:** Eritrean Conquest
**BiDAF Prediction:** Eritrean Conquest
**Analysis:** Similar to the previous example, both are confused by the additional sentence because the answer is obscured in the context. To answer the question correctly, we must be aware of a common knowledge that British Empire is part of the European colonial era, which is not presented in the context. Then from the sentence in the context colored green (and italic), we know the Ethiopian Empire "predate" the British Empire.

**ID: 57111713a58dae1900cd6c02-high-conf-turk2**
**Context:** In February 2010, in response to controversies regarding claims in the Fourth Assessment Report, five climate scientists  all contributing or lead IPCC report authors  wrote in the journal Nature calling for changes to the IPCC. They suggested a range of new organizational options, from tightening the selection of lead authors and contributors, to dumping it in favor of a small permanent body, or even turning the whole climate science assessment process into a moderated "living" Wikipedia-IPCC. *Other recommendations included that the panel employ a full-time staff and remove government oversight from its processes to avoid political interference*. It was suggested that the panel learn to avoid nonpolitical problems.

**Question:** How was it suggested that the IPCC avoid political problems?
**Answer:** remove government oversight from its processes

**FusionNet Prediction:** the panel employ a full-time staff and remove government oversight from its processes
**BiDAF Prediction:** the panel employ a full-time staff and remove government oversight from its processes
**Analysis:** In this example, both BiDAF and FusionNet are not confused by the added sentence. However, the prediction by both model are not precise enough. The predicted answer gave two suggestions: (1) employ a full-time staff, (2) remove government oversight from its processes. Only the second one is suggested to avoid political problems. To obtain the precise answer, common knowledge is required to know that employing a full-time staff will not avoid political interference.

**ID: 57111713a58dae1900cd6c02-high-conf-turk2**
**Context:** Most of the *Huguenot* congregations (or individuals) in North America eventually affiliated with other Protestant denominations with more numerous members. The *Huguenots* adapted quickly and *often married outside their immediate French communities*, which led to their assimilation. *Their descendants* in many families continued to use French first names and surnames for their children well into the nineteenth century. Assimilated, the French made numerous contributions to United States economic life, especially as merchants and artisans in the late Colonial and early Federal periods. *For example, E.I. du Pont,* a former student of Lavoisier, *established the Eleutherian gunpowder mills*. Westinghouse was one prominent Neptune arms manufacturer.

**Question:** Who was one prominent Huguenot-descended arms manufacturer?
**Answer:** E.I. du Pont

**FusionNet Prediction:** Westinghouse
**BiDAF Prediction:** Westinghouse
**Analysis:** This question requires both common knowledge and an understanding of the theme in the whole context to answer the question accurately. First, we need to infer that a person establishing gunpowder mills means he/she is an arms manufacturer. Furthermore, in order to relate E.I. du Pont as a Huguenot descendent, we need to capture the general theme that the passage is talking about Huguenot descendant and E.I. du Pont serves as an example.

# G   MULTI-LEVEL ATTENTION VISUALIZATION

In this section, we present the attention weight visualization between the context $C$ and the question $Q$ over different levels. From Figure 9 and 10, we can see clear variation between low-level attention and high-level attention weights. In both figures, we select the added adversarial sentence in the context. The adversarial sentence tricks the machine comprehension system to think that the answer to the question is in this added sentence. If only the high-level attention is considered (which is common in most previous architectures), we can see from the high-level attention map in the right hand side of Figure 9 that the added sentence

"The proclamation of the Central Park abolished protestantism in Belgium"

matches well with the question

"What proclamation abolished protestantism in France?"

This is because "Belgium" and "France" are similar European countries. Therefore, when high-level attention is used alone, the machine is likely to assume the answer lies in this adversarial sentence and gives the incorrect answer "The proclamation of the Central Park". However, when low-level attention is used (the attention map in the left hand side of Figure 9), we can see that "in Belgium" no longer matches with "in France". Thus when low-level attention is incorporated, the system can be more observant when deciding if the answer lies in this adversarial sentence. Similar observation is also evident in Figure 10. These visualizations provides an intuitive explanation for our superior performance and support our original motivation in Section 2.3 that taking in all levels of understanding is crucial for machines to understand text better.

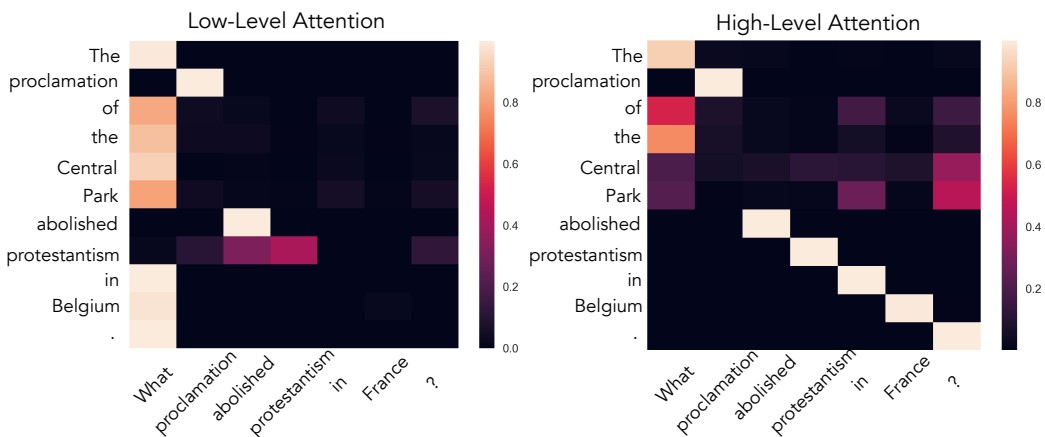

Figure 9: Multi-level Attention visualization between the added adversarial sentence and the question $Q$ on an article about Protestant Reformation.

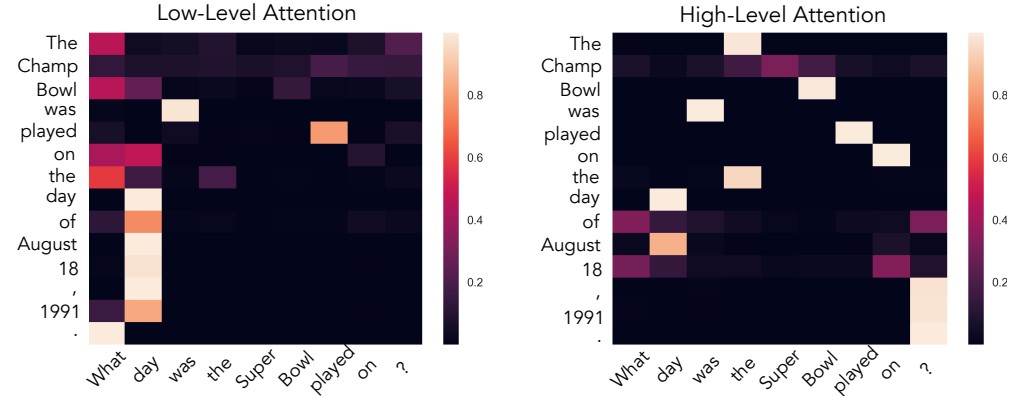

Figure 10: Multi-level attention visualization between the added adversarial sentence and the question $Q$ on an article about Super Bowl.

