# OpenReview forum: "FusionNet: Fusing via Fully-aware Attention with Application to Machine Comprehension"
_ICLR.cc/2018/Conference — Accept (Poster)_

### Official Review · AnonReviewer3 · 2017-11-25
**Nice analysis of literature, interesting model and good results, but a little lack of substance**

**Rating:** 7
**Confidence:** 5

**Review:**

The paper first analyzes recent works in machine reading comprehension (largely centered around SQuAD), and mentions their common trait that the attention is not "fully-aware" of all levels of abstraction, e.g. word-level, phrase-level, etc. In turn, the paper proposes a model that performs attention at all levels of abstraction, which achieves the state of the art in SQuAD. They also propose an attention mechanism that works better than others (Symmetric + ReLU).

Strengths:
- The paper is well-written and clear.
- I really liked Table 1 and Figure 2; it nicely summarizes recent work in the field.
- The multi-level attention is novel and indeed seems to work, with convincing ablations.
- Nice engineering achievement, reaching the top of the leaderboard (in early October).


Weaknesses:
- The paper is long (10 pages) but relatively lacks substances. Ideally, I would want to see the visualization of the attention at each level (i.e. how they differ across the levels) and also possibly this model tested on another dataset (e.g. TriviaQA).
- The authors claim that the symmetric + ReLU is novel, but  I think this is basically equivalent to bilinear attention [1] after fully connected layer with activation, which seems quite standard. Still useful to know that this works better, so would recommend to tone down a bit regarding the paper's contribution.


Minor:
- Probably figure 4 can be drawn better. Not easy to understand nor concrete.
- Section 3.2 GRU citation should be Cho et al. [2].


Questions:
- Contextualized embedding seems to give a lot of improvement in other works too. Could you perform ablation without contextualized embedding (CoVe)?


Reference:
[1] Luong et al. Effective Approaches to Attention-based Neural Machine Translation. EMNLP 2015.
[2] Cho et al. Learning Phrase Representations using RNN Encoder-Decoder for Statistical Machine Translation. EMNLP 2014.

---

> ### Author Response · Authors · 2017-12-06
> **Response for Reviewer 3's Review**
>
> Thank you so much for your thoughtful review!
>
> We acknowledged that our paper is quite long since we hope to best clarify the high-level concepts and the low-level details for fully-aware attention and our model, FusionNet. We are willing to add the visualization of multi-level attention weights in the appendix and will do so during the revision period. To support the generalization of our model, we have also tested on two adversarial datasets in addition to SQuAD and showed significant improvement. We are also working on extending to other datasets such as TriviaQA. While most machine comprehension models operate at paragraph-level (including ours), TriviaQA requires processing document-level input.  For example, the current state-of-the-art method on TriviaQA [1] uses a pipelined approach where machine comprehension model is only a part of the pipeline. Hence, we think a more in-depth study is needed to give a solid comparison on TriviaQA.
>
> We agree that the symmetric formulation is only a slightly modified version of standard multiplicative attention. However, during our research study on various architectures, incorporating history-of-word in attention score calculation only yields marginal improvements when existing attention formulations are used. On the other hand, when the slightly modified symmetric form is used, fully-aware attention becomes substantially better than normal attention. In the paper, we emphasized the importance of the symmetric attention form in the hope of the future researchers to utilize fully-aware attention better. We have rewritten our manuscript to stress the identification of the proper attention function and give less focus on the proposition of novel formulation. Thank you for pointing this out.
>
> Thank you for the question! CoVe is also helpful in our model. We have conducted some additional ablation study regarding input vectors based on your question.
>
> Performance on Dev set is shown (EM / F1):
> FusionNet : 75.3 / 83.6
> FusionNet without CoVe (original setting) : 74.1 / 82.5
> FusionNet without CoVe (embedding dropout=0.3) : 73.7 / 82.4
> FusionNet with GloVe fixed: 75.0 / 83.2
> =====
> Best documented number [3]: 72.1 / 81.6
>
> Note that we have optimized the hyperparameters in FusionNet with CoVe included. We have also tried our best to simplify the model when CoVe is available since we believe a simpler model is a better model. Therefore it is not very suitable to directly compare the ablation result with other models without CoVe. For example, we did not include character embedding (giving 2% improvement in BiDAF [4]) or multi-hop reasoning (giving 1% improvement in Reinforced Mnemonic Reader [3]) in our model.
>
> Minor:
> - We will try our best to improve figure 4. It will be very kind of you if you could give us some suggestions.
> - Thank you for pointing out the typo in GRU citation.
>
> Reference:
> [1] Clark, Christopher, and Matt Gardner. "Simple and Effective Multi-Paragraph Reading Comprehension." arXiv preprint arXiv:1710.10723 (2017)
> [2] Jia, Robin, and Percy Liang. "Adversarial examples for evaluating reading comprehension systems." EMNLP (2017).
> [3] Minghao Hu, Yuxing Peng, and Xipeng Qiu. "Reinforced Mnemonic Reader for Machine Comprehension." arXiv preprint arXiv:1705.02798 (2017).
> [4] Minjoon Seo, et al. "Bidirectional attention flow for machine comprehension." ICLR (2017).

---

### Official Review · AnonReviewer1 · 2017-12-02
**state of the art on SQuAD with FusionNet**

**Rating:** 8
**Confidence:** 3

**Review:**

The primary intellectual point the authors make is that previous networks for machine comprehension are not fully attentive. That is, they do not provide attention on all possible layers on abstraction such as the word-level and the phrase-level. The network proposed here, FusionHet, fixes problem. Importantly, the model achieves state-of-the-art performance of the SQuAD dataset.

The paper is very well-written and easy to follow. I found the architecture very intuitively laid out, even though this is not my area of expertise. Moreover, I found the figures very helpful -- the authors clearly took a lot of time into clearly depicting their work! What most impressed me, however, was the literature review. Perhaps this is facilitated by the SQuAD leaderboard, which makes it simple to list related work. Nevertheless, I am not used to seeing comparison to as many recent systems as are presented in Table 2.

All in all, it is difficult not to highly recommend an architecture that achieves state-of-the-art results on such a popular dataset.

---

> ### Author Response · Authors · 2017-12-09
> **Response for Reviewer 1's Review**
>
> Thank you so much for your encouraging review!

---

### Official Review · AnonReviewer2 · 2017-12-04
**Revisions have improved the paper**

**Rating:** 7
**Confidence:** 4

**Review:**

(Score before author revision: 4)
(Score after author revision: 7)

I think the authors have taken both the feedback of reviewers as well as anonymous commenters thoroughly into account, running several ablations as well as reporting nice results on an entirely new dataset (MultiNLI) where they show how their multi level fusion mechanism improves a baseline significantly. I think this is nice since it shows how their mechanism helps on two different tasks (question answering and natural language inference).

Therefore I would now support accepting this paper.

------------(Original review below) -----------------------

The authors present an enhancement to the attention mechanism called "multi-level fusion" that they then incorporate into a reading comprehension system. It basically takes into account a richer context of the word at different levels in the neural net to compute various attention scores.

i.e. the authors form a vector "HoW" (called history of the word), that is defined as a concatenation of several vectors:

HoW_i = [g_i, c_i, h_i^l, h_i^h]

where g_i = glove embeddings, c_i = COVE embeddings (McCann et al. 2017), and h_i^l and h_i^h are different LSTM states for that word.

The attention score is then a function of these concatenated vectors i.e. \alpha_{ij} = \exp(S(HoW_i^C, HoW_j^Q))

Results on SQuAD show a small gain in accuracy (75.7->76.0 Exact Match). The gains on the adversarial set are larger but that is because some of the higher performing, more recent baselines don't seem to have adversarial numbers.

The authors also compare various attention functions (Table 5) showing a particularone (Symmetric + ReLU) works the best.

Comments:

-I feel overall the contribution is not very novel.  The general neural architecture that the authors propose in Section 3 is generally quite similar to the large number of neural architectures developed for this dataset (e.g. some combination of attention between question/context and LSTMs over question/context). The only novelty is these "HoW" inputs to the extra attention mechanism that takes a richer word representation into account.

-I feel the model is seems overly complicated for the small gain (i.e. 75.7->76.0 Exact Match), especially on a relatively exhausted dataset (SQuAD) that is known to have lots of pecularities (see anonymous comment below). It is possible the gains just come from having more parameters.

-The authors (on page 6) claim that that by running attention multiple times with different parameters but different inputs (i.e. \alpha_{ij}^l, \alpha_{ij}^h, \alpha_{ij}^u) it will learn to attend to "different regions for different level". However, there is nothing enforcing this and the gains just probably come from having more parameters/complexity.

---

> ### Author Response · Authors · 2017-12-05
> **Response for Reviewer 2's Review**
>
> Thank you for your review!
>
> 1. Our improvement over best model published is actually ~3% in Exact Match (73.2 -> 76.0). We understand that from our Table 2, it seems FusionNet only improve the best "published" model (R-net) by EM 0.3 (single model). We apologize for not writing this part clear and have updated our paper accordingly. If you look into the ACL2017 paper of R-net [1] or the recent technical report (http://aka.ms/rnet), you will find that the best-published version of R-net only achieves EM: 72.3, F1: 80.7. It is lower than our model by near 4% in EM. It is because the authors of R-net have been designing new models without publishing it while using the same model name (R-net) on SQuAD leaderboard. At the time of ICLR2018 submission, the best-published model is Reinforced Mnemonic Reader [2] (https://arxiv.org/pdf/1705.02798.pdf), which achieved EM: 73.2, F1: 81.8, 1% higher than published version of R-net. Reinforced Mnemonic Reader proposed feature-rich encoder, semantic fusion unit, iterative interactive-aligning self-aligning, multihop memory-based answer pointer, and a reinforcement learning technique to achieve their high performance. On the other hand, utilizing our simple "HoW" attention mechanism, FusionNet obtained a decent performance (EM: 76.0, F1: 83.9) on original SQuAD with a relatively simplistic model. For example, in Table 6, by changing S(h_i^C, h_j^Q) to S(HoW_i^C, HoW_j^Q) in a vanilla model (encoder + single-level attention), we observed +8% improvement and achieved EM: 73.3, F1: 81.4 on the dev set. (best documented number on dev set: 72.1 / 81.6)
>
> 2. In our paper, we only compare with the official results on adversarial dataset shown in this year EMNLP paper [3]. Adversarial evaluation of more recent higher performing models can be found in a website maintained by the author (Robin Jia). And we still significantly outperform these recent state-of-the-art methods.
> https://worksheets.codalab.org/worksheets/0x77ca15a1fc684303b6a8292ed2167fa9/
> For example, Robin Jia has compared with another state-of-the-art model DCN+, which is also submitted to ICLR2018. On the AddSent dataset, DCN+ achieved F1: 44.5, and on the AddOneSent dataset, DCN+ achieved F1: 54.3. The results are comparable to the previous state-of-the-art on the adversarial datasets. But FusionNet is +6% higher than DCN+ on both datasets. We attribute this significant gain to the proposed HoW attention, which can very easily incorporate into other models. We are excited to share this simple idea with the community to improve machines in better understanding texts.
>
> 3. SQuAD is a very competitive dataset, so it is unlikely that the significant gain we have (+3% over best-documented models, +5% in adversarial datasets) comes from giving the model more parameters. Furthermore, existing models can always incorporate more parameters if it helps. For example, in the high-performing Reinforced Mnemonic Reader, they can increase the number of iterations in iterative aligning or increase the hidden size in LSTM. Additionally, in our Table 6, we have compared FA All-level and FA Multi-level. FA All-level uses the same attention weight for different levels and fuses all level of representation (including input vector). FA All-level has more parameters than FA Multi-level but performs 2% worse. Based on Reviewer3's comment, we will also include visualization to show that multi-level attention will learn to attend to "different regions for different levels."
>
> References:
> [1] Wenhui Wang, Nan Yang, Furu Wei, Baobao Chang, and Ming Zhou. "Gated self-matching networks for reading comprehension and question answering." ACL (2017).
> [2] Minghao Hu, Yuxing Peng, and Xipeng Qiu. "Reinforced Mnemonic Reader for Machine Comprehension." arXiv preprint arXiv:1705.02798 (2017).
> [3] Robin Jia, and Percy Liang. "Adversarial examples for evaluating reading comprehension systems." EMNLP (2017).

---

### Public Comment · (anonymous) · 2017-11-10
**Nice.**

Nice simple model.

---

> ### Author Response · Authors · 2017-11-18
> **Re: Nice.**
>
> Thank you for your compliment!

---

### Public Comment · (anonymous) · 2017-11-18
**Reproductive Study - Training Parameters**

I set up a reproduction experiment for which I need a little clarification on the following.

- Which components are used/set up without dropout?
- Attention dimensions for the various fusions present.

---

> ### Author Response · Authors · 2017-11-18
> **Training Parameters**
>
> Thank you for being interested in reproducing our work!
>
> - Which components are used without dropout?
> Dropout is applied before every linear transform, including the input for each layer of LSTM and Attention. For fast implementation, we do not use hidden state dropout in LSTM. Also, an additional dropout is applied after the GloVe and CoVe embedding layer.
> The dropout is shared across time step (i.e., variational dropout). And different linear layers use different dropout masks.
>
> - Attention dimensions for the various fusions present.
> For all the fully-aware attention S(HoW_i, HoW_j), we used an attention dimension k = 250 (the same as the output size of BiLSTM).

---

### Public Comment · (anonymous) · 2017-12-03
**Only SQuAD Evaluation!?**

I noticed that you only evaluate against SQuAD which is known to be a bad dataset for evaluating machine comprehension. It has only short documents and most of the answers are easily extractable. This is a bit troubling especially given that there are plenty of good and much more complex datasets out there, e.g., TriviaQA, NewsQA, just to mention a few. It feels like we are totally overfitting on a simple dataset. Would it be possible to also provide results on one of those, otherwise it is really hard to judge whether there is indeed any significant improvement. I think this is a big issue.

---

> ### Author Response · Authors · 2017-12-05
> **Additional Datasets are Evaluated on FusionNet**
>
> Hi!
>
> We are sorry to hear about your view on SQuAD. However, in our opinion, SQuAD is one of the best machine comprehension datasets currently available. The community also has a consensus on this. SQuAD has received the best resource paper award at EMNLP, and there has been active participation from many institutions (Salesforce, Google, Stanford, AI2, Microsoft, CMU, FAIR, HIT and iFLYTEK). Furthermore, many recent papers have shown that the performance on other datasets, such as TriviaQA, often supports the improvement on SQuAD. For example, in the comparison of [1], the F1 performances of three high-performing models (BiDAF, MEMEN, M-reader+RL) on SQuAD are BIDAF: 81.5, MEMEN: 82.7 (+1.2), M-reader+RL: 84.9 (+3.4). On the TriviaQA Web domain verified setting, the F1 performance becomes BIDAF: 55.8, MEMEN: 57.6 (+1.8), M-reader+RL: 61.5 (+5.7), which highly authenticate the improvement seen in SQuAD.
>
> Nonetheless, we agree with your concern that a high performance on SQuAD may not always generalize to a different dataset. It is the exact motivation of recently proposed adversarial datasets [2], which has received this year EMNLP outstanding paper award. These adversarial datasets are very challenging. The accuracy of sixteen published models drops from an average of 75% F1 score on SQuAD to 36% on these datasets. Furthermore, many high-performing models on SQuAD performs particularly bad on these adversarial datasets.
>
> To verify the generalization of our model in addition to SQuAD, we have also evaluated on two challenging adversarial datasets proposed in [2], AddSent and AddOneSent. From our paper, you can see that FusionNet not only performs well on SQuAD but also achieves a significant improvement (+5%) on these adversarial datasets over state-of-the-art methods.
>
> References:
> [1] Minghao Hu, Yuxing Peng, and Xipeng Qiu. "Reinforced Mnemonic Reader for Machine Comprehension." arXiv preprint arXiv:1705.02798 (2017).
> [2] Robin Jia, and Percy Liang. "Adversarial examples for evaluating reading comprehension systems." EMNLP (2017).

---

> > ### Public Comment · (anonymous) · 2017-12-05
> > **RE**
> >
> > Well, a lot of dataset papers and papers have shown that SQuAD is overly simple, due to limited context size and a lot of paraphrases. It requires only a rather simple heuristic to solve it. So not the entire community agrees on that.
> >
> > The improvements on adversarial datasets are also not convincing by themselve, because (for some reason) you merely evaluate the ensemble against it, which is extremely large (more than 30 models), much larger (please correct me if I am wrong) than other ensembles. Since it is known that ensembling makes systems more robust against such attacks, this is not really comparable. Showing that a single model is more robust might help make this claim stronger though.

---

> > > ### Author Response · Authors · 2017-12-06
> > > **RE**
> > >
> > > Hi!
> > >
> > > We don't think there is any "simple heuristic" that could "solve" SQuAD nor does any literature mention such a comment. We presume that you might be referring to FastQA [1] (EM / F1 = 68.4 / 77.1) and DrQA [2] (EM / F1 = 70.0 / 79.0). Both are great models that are conceptually simple, yet they contain nontrivial uses of LSTM and attention. They also perform well on machine comprehension tasks other than SQuAD. We highly respect these two models since a relatively simple model that performs well is an exceptionally great model. In the design of FusionNet, we have also taken this philosophy into account and try our best to simplify our model.
> > >
> > > Additionally, FusionNet have improved SQuAD performance to EM / F1 = 76.0 / 83.9. The improvement is significant (+6% in EM). We would not regard these model, including our FusionNet, to have solved SQuAD.
> > >
> > > The reason for showing only the ensemble performance on the adversarial datasets is solely due to the space constraint. Furthermore, for most models, the ensemble performs better or on par with the single model on adversarial datasets.
> > >
> > > We understand your concern since most ensemble contains 10~20 models. We will train a 10-model ensemble of FusionNet and evaluate this smaller ensemble on adversarial datasets.
> > >
> > > References:
> > > [1] Dirk Weissenborn, Georg Wiese, Laura Seiffe. "Making Neural QA as Simple as Possible but not Simpler." CoNLL (2017).
> > > [2] Danqi Chen, et al. "Reading Wikipedia to Answer Open-Domain Questions." ACL (2017).

---

> > > > ### Public Comment · (anonymous) · 2017-12-06
> > > > **RE**
> > > >
> > > > The mentioned models do not contain attention at all. They compute a weighted question representation that is all. So these model clearly indicate that much of SQuAD is more or less trivial. However, I am not saying that your model is not a contribution, only that it should be evaluated on another dataset since there are quite a few now. Otherwise, your improvements can stem from overfitting your architecture to SQuAD.
> > > >
> > > > I think having less comparable number of models in the ensemble is necessary to get the clear picture. This evaluation should also be done without CoVe, to be really comparable.

---

> > > > > ### Public Comment · (anonymous) · 2017-12-09
> > > > > **Totally agree doing an evaluation without CoVe**
> > > > >
> > > > > The authors should include results of evaluating their model without CoVe in the submitted paper to avoid a confusion whether their complex model is good a by feature engineering or not.

---

### Public Comment · (anonymous) · 2017-12-04
**Ablation without CoVE? SotA through better word embeddings?**

Hi, most models this paper compares to are trained with GloVe embeddings but you only show results with CoVe (if I am not mistaken). I feel like this model is only able to achieve SotA because it uses CoVe and not because of the additional extensions.  Is this correct? Do you have an ablation for that?

---

> ### Author Response · Authors · 2017-12-06
> **Re: Ablation Study on Input Vectors**
>
> Hi!
>
> We would kindly disagree with your statement. First of all, every model has it's unique input embeddings. BiDAF [1] uses the output of highway network that combines GloVe and Char embedding as the input vector. Reinforced Mnemonic Reader [2] proposes Feature-rich Encoder which includes additional features such as query category. It is not suitable to criticize that BiDAF achieved a previous SotA solely because of the Char embedding. The neural architecture as a whole is what delivers the SotA performance.
>
> Nevertheless, we agree that including an ablation study on input vectors can help the community better understand the model. Here is the ablation study for FusionNet.
>
> Performance on Dev set is shown (EM / F1):
> FusionNet : 75.3 / 83.6
> FusionNet without CoVe (original setting) : 74.1 / 82.5
> FusionNet without CoVe (embedding dropout=0.3) : 73.7 / 82.4
> FusionNet with GloVe fixed: 75.0 / 83.2
> =====
> Best documented number [2]: 72.1 / 81.6
>
> As you can see, CoVe is indeed helpful in FusionNet. However, the ablated performance is still higher than the best model published (Reinforced Mnemonic Reader) by 2% in EM.
>
> Nevertheless, it is not appropriate to directly compare our ablation result with other models without CoVe. We have optimized the hyperparameters in FusionNet when CoVe is present, and we have tried our best to simplify FusionNet when CoVe is included due to our belief that a simple model is a better model. For example, we did not include multi-hop reasoning [2] (giving reinforced mnemonic reader 1% improvement), character embedding (giving BiDAF 2% improvement) ... etc.
>
> The core of FusionNet is the Fully-Aware Attention. We have found this simple enhancement to be exceptionally advantageous for machine comprehension. We sincerely hope the community can benefit from this simple but powerful idea.
>
> References:
> [1] Minjoon Seo, et al. "Bidirectional attention flow for machine comprehension." ICLR (2017).
> [2] Minghao Hu, Yuxing Peng, and Xipeng Qiu. "Reinforced Mnemonic Reader for Machine Comprehension." arXiv preprint arXiv:1705.02798 (2017).

---

> > ### Public Comment · (anonymous) · 2017-12-06
> > **RE**
> >
> > Thanks for the clarification! It is nice to see that the model is still (slightly) better, but given the complexity this is not surprising. Overall the results are less impressive now. I also do not really buy the term fully aware attention. It is just attention on multiple levels, an engineering trick.

---

> > > ### Author Response · Authors · 2017-12-06
> > > **Re**
> > >
> > > In our opinion, FusionNet is architecturally simpler than previous state-of-the-art methods (such as Reinforced Mnemonic Reader, which utilizes feature-rich encoder, semantic fusion unit, iterative interactive-aligning self-aligning, multihop memory-based answer pointer, and a reinforcement learning technique to achieve their great performance) in many ways while performing better. We are sorry to hear your opinion.
> > >
> > > Still, we hope this simple and powerful technique can help in your future application of machine comprehension!

---

> > > > ### Public Comment · (anonymous) · 2017-12-06
> > > > **RE**
> > > >
> > > > I agree, it is simpler than that model, but not at all simpler than, for instance, r-net which performs almost as well. Anyway, simplicity is just a personal preference for me until something more complex is able to make a significant improvement on *many* datasets. I believe we create complex networks in abundance, but it turns out that fair comparisons almost always show that they do not give a significant gain (both empirically and wrt. research insights). This, however, is my very subjective opinion.

---

### Author Response · Authors · 2018-01-03
**Paper Revision**

We have conducted additional experiments based on Reviewer 3's thoughtful comments and included the results in our paper.

(1) Additional ablation study on input vectors. (Appendix C)

The following is a summary of our additional experimental results. The experiment shows that FusionNet, with or without CoVe, single or ensemble, all yield clear improvement over previous state-of-the-art models on all 3 machine comprehension datasets.

=== SQuAD (Dev EM) ===
> FusionNet: 75.3 (+3.2)
> FusionNet (without CoVe): 74.1 (+2.0)
Previous SotA [1]: 72.1

=== AddSent (Dev EM) ===
> FusionNet (Single): 45.6 (+4.9)
> FusionNet (Single, without CoVe): 47.4 (+6.7)
> FusionNet (Ensemble): 46.2 (+5.5)
Previous SotA [2] (Ensemble): 40.7

=== AddOneSent (Dev EM) ===
> FusionNet (Single): 54.8 (+6.1)
> FusionNet (Single, without CoVe): 55.2 (+6.5)
> FusionNet (Ensemble): 54.7 (+6.0)
Previous SotA [2] (Ensemble): 48.7

(2) Application to Natural Language Inference. (Appendix D)

FusionNet is an improved attention mechanism that can be easily applied to attention-based models. Here, we consider a different task on natural language inference (NLI). We focus on MultiNLI [3], a recent NLI corpus. MultiNLI is designed to be more challenging than SNLI [4] since many models already outperform human performance (87.7%) on SNLI. A state-of-the-art model for NLI is ESIM [5], which performs 88.0% on SNLI and 72.3% (in-domain), 72.1% (cross-domain) on MultiNLI [3]. We implemented a version of ESIM in PyTorch and improved ESIM with our proposed fully-aware multi-level attention mechanisms. For fair comparisons, we reduce the hidden size after adding our enhancements, so the parameter size after attention enhancement is less than or similar to ESIM with standard attention. A summary of the result is shown below. Experiments on natural language inference conform with our observed improvements in machine comprehension tasks.

=== MultiNLI (Dev Cross-domain / In-domain) ===
Our ESIM (d = 300): 73.9 / 73.7
Our ESIM + fully-aware (d = 250): 77.3 / 76.5
Our ESIM + fully-aware + multi-level (d = 250): 78.4 / 78.2

(3) Multi-level attention visualization. (Appendix G)

In Appendix G, we included multi-level attention visualization and a qualitative analysis of the attention variations between low-level and high-level. These visualizations support our original motivation and provide an intuitive explanation for our superior performance, especially on the adversarial datasets.

References:
[1]: Minghao Hu, Yuxing Peng, and Xipeng Qiu. "Reinforced Mnemonic Reader for Machine Comprehension." arXiv preprint arXiv:1705.02798 (2017).
[2]: Robin Jia, and Percy Liang. "Adversarial examples for evaluating reading comprehension systems." EMNLP (2017).
[3]: Adina Williams, Nikita Nangia, and Samuel R. Bowman. "A broad-coverage challenge corpus for sentence understanding through inference." arXiv preprint arXiv:1704.05426 (2017).
[4]: Samuel R. Bowman et al. "A large annotated corpus for learning natural language inference." EMNLP (2015).
[5]: Qian Chen et al. "Enhancing and combining sequential and tree lstm for natural language inference." ACL (2017).

---

### Public Comment · (anonymous) · 2018-01-20
**Training vs dev ratio**

The performance of the model on SQuAD dataset is impressive. In addition to the performance on the test set, we are also interested in the sample complexity of the proposed model. Currently, the SQuAD dataset splits the collection of passages into a training set, a development set, and a test set in a ratio of 80%:10%:10% where the test set is not released. Given the released training and dev set, we are wondering what would happen if we split the data in a different ratio, for example, 50% for training and the rest 50% for dev. We will really appreciate it if the authors can report the model performance (on training/dev respectively) under this scenario.

---

### Public Comment · (anonymous) · 2018-01-23
**Question on Ensemble model and Experimental Setup**

I loved reading your paper. Very well written. Great work!
I would like to know more about your ensemble model. You have mentioned that your ensemble contains 39 models. Can you please comment on what these models are?

How long did the training take during your experiments? What is the batch size? And GPU memory requirements? Also, do you plan to open-source your code?

---

> ### Author Response · Authors · 2018-02-04
> **Re: Question on Ensemble model and Experimental Setup**
>
> It is great to hear your positive comment!
>
> In the paper, we used 31 models consisting of different random initialization. On a Titan Xp GPU, each epoch takes ~20 minutes. A plot of performance versus epoch can be found in Appendix A. The batch size is 32. A GPU with 12 GB memory is enough. We only use about 8 GB under PyTorch.
>
> An implementation of FusionNet under PyTorch can be found here.
> https://github.com/momohuang/FusionNet-NLI

---

### Public Comment · (anonymous) · 2018-02-21
**Query about the training procedure**

The work is very well presented. I have one question. Do you train the embedding matrix of PoS and NER tagging?

---

> ### Author Response · Authors · 2018-04-11
> **Re: Query about the training procedure**
>
> Hi,
>
> Thank you! Yes, we did train the embedding matrix of PoS and NER embedding.

---

### Decision · Program_Chairs · 2018-01-29
**ICLR 2018 Conference Acceptance Decision**

**Decision:**

Accept (Poster)

**Comment:**

State-of-the-art results on Squad (at least at time of submission) with a nice model. Authors have since applied the model to additional tasks (SNLI). Good discussion with reviewers, well written submission and all reviewers suggest acceptance.